# CaMKII suppresses proteotoxicity by phosphorylating BAG3 in response to proteasomal dysfunction

Chenliang Zhang [ID][1,3], Huanji Xu[2,3], Qiulin Tang[1], Yichun Duan[2], Hongwei Xia[1], Huixi Huang[1], Di Ye[2] & Feng Bi [ID][1,2 ✉]

## Abstract

**Protein quality control serves as the primary defense mechanism for cells against proteotoxicity induced by proteasome dysfunction. While cells can limit the build-up of ubiquitinated misfolded proteins during proteasome inhibition, the precise mechanism is unclear. Here, we find that protein kinase Ca$^{2+}$/Calmodulin (CaM)-dependent protein kinase II (CaMKII) maintains proteostasis during proteasome inhibition. We show that proteasome inhibition activates CaMKII, which phosphorylates B-cell lymphoma 2 (Bcl-2)-associated athanogene 3 (BAG3) at residues S173, S377, and S386. Phosphorylated BAG3 activates the heme-regulated inhibitor (HRI)- eukaryotic initiation factor-2α (eIF2α) signaling pathway, suppressing protein synthesis and the production of aggregated ubiquitinated misfolded proteins, ultimately mitigating the proteotoxic crisis. Inhibition of CaMKII exacerbates the accumulation of aggregated misfolded proteins and paraptosis induced by proteasome inhibitors. Based on these findings, we validate that combined targeting of proteasome and CaMKII accelerates tumor cell death and enhances the efficacy of proteasome inhibitors in tumor treatment. Our data unveil a new proteasomal inhibition-induced misfolded protein quality control mechanism and propose a novel therapeutic intervention for proteasome inhibitor-mediated tumor treatment.**

**Keywords** Proteasome Inhibition; CaMKII; BAG3; HRI; Paraptosis
**Subject Categories** Cancer; Post-translational Modifications & Proteolysis; Signal Transduction

## Introduction

Proteasome activity insufficiency can lead to pathological biological processes, such as accumulating ubiquitinated misfolded proteins, endoplasmic reticulum stress, oxidative stress, and cell damage (Paul, 2008). Among them, aggregation of ubiquitinated misfolded proteins is an essential cellular biological phenomenon (Goldberg, 2003; Ciechanover, 2005). During proteasomal inhibition, misfolded proteins, often polyubiquitinated, are assembled into micro-aggregates. These aggregates are then transported along microtubules to the microtubule organizing center (MTOC), where they loosely congregate to form perinuclear aggresomes (Johnston et al, 1998; Kawaguchi et al, 2003). Although aggregation of ubiquitinated misfolded proteins is beneficial for cells to reduce proteotoxic stress in response to proteasome impairment, excessive accumulation will also enhance proteotoxicity and aggravate cell damage (Johnston et al, 1998; Takahashi et al, 2018; Zhang et al, 2022a, 2022b; Stefani et al, 2003).

B-cell lymphoma 2 (Bcl-2)-associated athanogene 3 (BAG3) is a critical regulatory protein in ubiquitinated misfolded protein aggregation and proteotoxicity reduction during proteasome inhibition (Klimek et al, 2017). As a multifunctional protein, BAG3 contains several characteristic structural motifs and domains: WW domain (tryptophan-tryptophan domain) at the N-terminus, which mediates the interaction with proline-rich repeats of proteins such as Large tumour suppressor 1/2 (LATS1/2), tuberous sclerosis complex 1 (TSC1); two IPV motifs (isoleucine–proline–valine motif), which mainly determine the binding of BAG3 to small molecule heat-shock proteins (HSPs), such as HSPB/68; a PxxP repeat motif (proline-rich region), which mediates the binding of BAG3 to Src homology 3 (SH3), such as PLC-γ and Dynein; and a highly conserved BAG domain in the C-terminus, which mediates the binding of BAG3 to HSP70, Bcl-2, and Heat-Shock Factor 1 (HSF1) (Stürner et al, 2017). As a co-chaperone, the primary function of BAG3 is regulating protein quality control and proteostasis through its BAG domain association with Hsc/Hsp70. Proteasome impairment can decrease the association of HSPs to BAG1, which mainly mediates the proteasomal degradation of ubiquitinated proteins, and promote HSPs-BAG3 binding. The BAG3/HSPs complex can facilitate the aggregation of ubiquitinated misfolded proteins (Meriin et al, 2018; Minoia et al, 2014; Sha et al, 2009; Gamerdinger et al, 2011;

[1]Division of Abdominal Tumor Multimodality Treatment, Department of Medical Oncology, Cancer Center and Laboratory of Molecular Targeted Therapy in Oncology, West China Hospital, Sichuan University, Chengdu, Sichuan Province, China. [2]Division of Abdominal Tumor Multimodality Treatment, Department of Medical Oncology, Cancer Center, West China Hospital, Sichuan University, Chengdu, Sichuan Province, China. [3]These authors contributed equally: Chenliang Zhang, Huanji Xu. ✉E-mail: bifeng@scu.edu.cn

Zhang et al, 2011; Zhang et al, 2022b). Recently, BAG3 has also been discovered to play a role in the regulation of eukaryotic initiation factor-2α (eIF2α)-associated integrated stress response, which can influence the level of protein synthesis in cells, in response to proteasome dysfunction (Carra, et al, 2009; Patel et al, 2022). However, the molecular mechanism remains elusive.

The Ca²⁺/Calmodulin (CaM)-dependent protein kinase II (CaMKII) form a family of evolutionarily conserved protein kinases that Ca²⁺/CaM can activate. This family consists of four members, CaMKIIα (*CAMK2A*), CaMKIIβ (*CAMK2B*), CaMKIIγ (*CAMK2G*), and CaMKIIδ (*CAMK2D*). All CaMKII subunits contain a highly conserved kinase domain at the N-terminus, an oligomerization domain (or Hub domain) at the C-terminus, an autoregulatory domain, and a linker domain in the middle (Coultrap et al, 2012; Gaertner et al, 2004; Fink et al, 2002). Although the kinase domain is highly conserved, there are variable transcription splices in the linker domain between the autoregulatory and oligomerization domains, resulting in the diversity of the proteins (Coultrap et al, 2012; Gaertner et al, 2004; Fink et al, 2002). In the inactive states, different CaMKII subunits copolymerize to form a hub-and-spoke structure through their oligomerization domains, in which their autoregulatory domain blocks the kinase domain (Gaertner et al, 2004; Fink et al, 2002). The binding of Ca²⁺/CaM to the autoregulatory segment releases the catalytic site of the kinase domain, consequently, mediates the autophosphorylation of neighboring CaMKII at T286 (CaMKIIα) or T287 (CaMKIIβ/γ/δ). This autophosphorylation can further improve the binding ability of CaMKII to the Ca²⁺/CaM and enhance the protein kinase activity (Coultrap et al, 2012; Gaertner et al, 2004; Fink et al, 2002; Tsujioka et al, 2023).

Paraptosis is a programmed cell death mode characterized by the endoplasmic reticulum (ER), mitochondria dilation, or both (Hanson et al, 2023; Lee et al, 2016). Although the molecular basis of paraptosis remains unclear, studies showed that paraptosis is accompanied by ubiquitinated protein accumulation, reactive oxygen species overload, ER stress aggravation, mitogen-activated protein kinase (MAPK) pathway activation, and Alix protein reduced levels (Hanson et al, 2023; Lee et al, 2016; Sperandio et al, 2004; Li et al, 2022; Rozpedek et al, 2016). Among them, the excessive accumulation of ubiquitinated proteins is the leading cause of paraptosis. Therefore, suppressing protein synthesis can effectively inhibit paraptosis (Hanson et al, 2023; Lee et al, 2016; Sperandio et al, 2004; Li et al, 2022; Rozpedek et al, 2016). Disturbing the regulatory network of ubiquitinated misfolded protein accumulation during proteasome inhibition aggravates paraptosis-like cell death (Yoon et al, 2010; Nedungadi et al, 2018; Lee et al, 2017), suggesting strict monitoring mechanisms for paraptosis in response to proteasome inhibition.

This study investigated the mechanism of CaMKII regulated the accumulation of aggregated misfolded proteins and the cell damage induced by proteasome inhibitors. The results showed that activated CaMKII induced by proteasome inhibition could phosphorylate BAG3 at S173/377/386, which mediates heme-regulated inhibitor (HRI)- eukaryotic initiation factor-2α (eIF2α) signaling activation, thereby inhibiting the synthesis of aggregated ubiquitinated proteins and reduces paraptosis. Based on the mechanism, we also showed that a combination of CaMKII inhibitors could enhance the anti-tumor activity of proteasome

inhibitors. Collectively, our results uncover a new mechanism of proteasomal inhibition-induced misfolded protein quality control. Moreover, we suggest a novel therapeutic strategy for proteasome inhibitor-mediated tumor treatment.

# Results

## CaMKII regulates the synthesis of aggregated ubiquitinated proteins in response to proteasome inhibition

To investigate the functions of CaMKII during proteasome inhibition, we first examined CaMKII activation by accessing the phosphorylation level of CaMKII T286/287. The results revealed that the proteasome inhibitors, MG132 or Bortezomib, significantly enhanced the phosphorylation of CaMKII T286/T287 in HEK293 and HeLa cells (Fig. 1A–D). This suggests that proteasomal inhibition promotes CaMKII activation. Next, the effect of CaMKII on the quality control of ubiquitinated misfolded proteins was investigated. We expressed shRNA plasmids targeting CaMKIIs in HEK293 and HeLa cells to knock down their expression levels (Fig. EV1A). We found that during proteasome inhibition, CaMKII knockdown significantly increased ubiquitinated protein accumulation in the NP-40-insoluble fractions (Fig. 1E), where protein aggregates accumulate as they cannot be solubilized by mild detergent (Kawaguchi et al, 2003; Zhang et al, 2022a; Minoia et al, 2014). Additionally, we examined ubiquitinated protein aggregation by immunostaining UB-K48, a well mark of proteasome inhibition-induced aggregates of ubiquitinated protein (Zhang et al, 2022a; Morrow et al, 2020), revealing that knockdown CaMKII aggravated ubiquitinated protein aggregation during proteasome inhibition (Fig. 1F,G). Moreover, KN-93, a CaMKII inhibitor, aggravated ubiquitinated protein accumulation (Fig. 1H), including K48- and K63-linked polyubiquitinated proteins (Fig. EV1B), in the NP-40-insoluble fractions induced by a proteasome inhibitor. Immunofluorescence staining also showed that CaMKII inhibition aggravated proteasome inhibition-induced ubiquitinated protein aggregation (Fig. 1I,J), including micro-aggregate accumulation and aggresome formation. Collectively, these results indicate that CaMKII regulates aggregated ubiquitinated protein accumulation in response to proteasome inhibition.

Subsequently, the autophagic degradation of ubiquitinated proteins was examined to understand whether aggregated protein accumulation caused by CaMKII inhibition is attributed to inefficient aggrephagy. The results revealed that the lysosomal inhibitor, Pep/E64d, did not affect proteasome inhibition-induced the accumulation of ubiquitinated protein in NP-40 insoluble fractions, in the presence or absence of KN-93 (Fig. EV1C), suggesting CaMKII does not regulate the autophagic degradation of ubiquitinated proteins during proteasome inhibition. Previous studies revealed that during proteasome inhibition, aggregated ubiquitinated proteins are mainly derived from newly translated misfolded proteins (Sung et al, 2016; Mimnaugh et al, 2004). Therefore, we speculate that CaMKII may suppress ubiquitinated misfolded protein production. We examined the protein synthesis by Click-iT HPG technology and found proteasome inhibitor decreased the rate of protein synthesis in cells, which was reversed by the KN-93 treatment (Fig. 1K). This suggests that CaMKII blocks the synthesis of misfolded proteins in response to proteasome inhibition. Moreover,

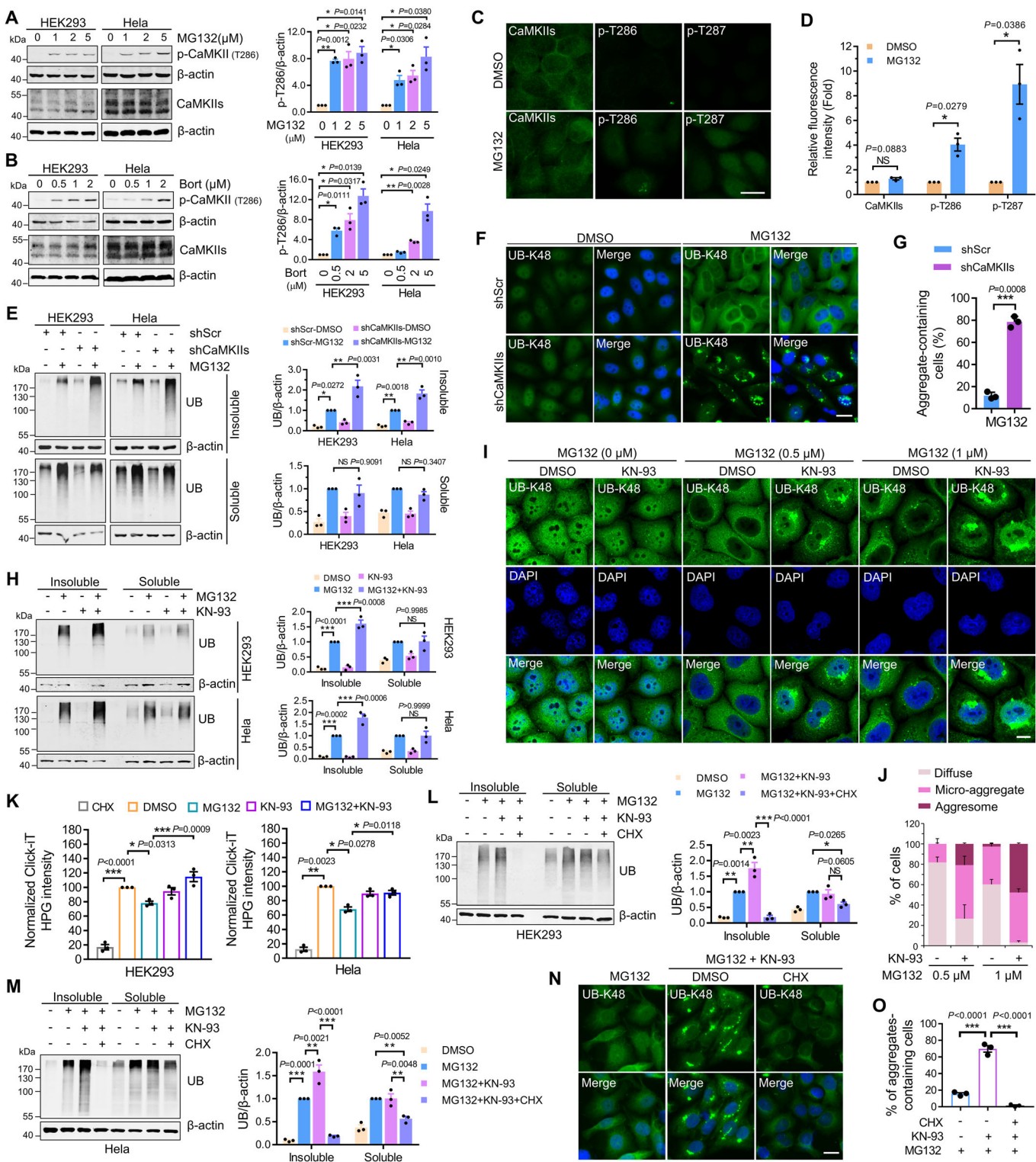

protein synthesis suppression using Cycloheximide (CHX) can effectively reverse aggregated ubiquitinated protein accumulation induced by CaMKII inhibitors (Fig. 1L–O). These results indicate that CaMKII could regulate aggregated ubiquitinated protein accumulation by suppressing misfolded protein synthesis during proteasome inhibition.

## Inhibition of CaMKII aggravates proteasome inhibitor-induced paraptosis

Excessive accumulation of aggregated proteins may enhance the cell damage induced by proteasome inhibition. Regarding cell morphology, CaMKIIs knockdown or inhibition significantly

**Figure 1. CaMKII regulates the synthesis of aggregated ubiquitinated proteins in response to proteasome inhibition.**

(A, B) Immunoblotting and quantification of p-CaMKII (T286) in HEK293 and HeLa cells treated with MG132 (A) or Bortezomib (Bort) (B) at indicated concentrations for 14 h. Data are mean ± SEM of biological replicates ($n = 3$). (C) Representative CaMKIIs, p-CaMKII (T286), and p-CaMKII (T287) staining images of HeLa cells treated with MG132 (1 μM) for 14 h. Scale bar: 20 μm. (D) Immunofluorescence-based measurement and quantification of the expression levels of indicated proteins shown in (C). Data are mean ± SEM of biological replicates ($n = 3$). (E) Immunoblotting and quantification of ubiquitin (UB) in NP-40-soluble and -insoluble fractions of HEK293 and HeLa cells transfected with indicated shRNA and treated with MG132 (1 μM) for 14 h. Data are mean ± SEM of biological replicates ($n = 3$). (F) Representative UB-K48 staining images of HeLa cells transfected with indicated shRNA and treated with MG132 (0.5 μM) for 14 h. Scale bar: 20 μm. (G) Quantitative analysis of results in (E). Data are mean ± SEM of biological replicates ($n = 3$). At least 50 cells were randomly selected from each group to score. (H) Immunoblotting and quantification of UB in NP-40-soluble and -insoluble fractions of HEK293 and HeLa cells treated with indicated drugs (MG132 (1 μM), KN-93 (10 μM)) for 14 h. Data are mean ± SEM of biological replicates ($n = 3$). (I) Representative UB-K48 staining images of HeLa cells treated with indicated drugs for 14 h. Scale bar: 20 μm. (J) Quantitative analysis of results in (I). Data are mean ± SEM of biological replicates ($n = 3$). At least 50 cells were randomly selected from each group to score. (K) Protein synthesis was detected using a Click-iT HPG method in HEK293 and HeLa cells treated indicated drugs (MG132 (1 μM), KN-93 (10 μM)) for 14 h. Data are mean ± SEM of biological replicates ($n = 3$). (L, M) Immunoblotting and quantification of UB in NP-40-soluble and -insoluble fractions of HEK293 (L) and HeLa (M) cells treated with indicated drugs (MG132 (1 μM), KN-93 (10 μM), Cycloheximide (CHX, 25 μg/ml)) for 14 h. Data are mean ± SEM of biological replicates ($n = 3$). (N) Representative UB-K48 staining images of HeLa cells treated with indicated drugs as in (MG132 (0.5 μM), KN-93 (10 μM), Cycloheximide (CHX, 25 μg/ml)) for 14 h. Scale bar: 20 μm. (O) Quantitative analysis of results in (N). Data are mean ± SEM of biological replicates ($n = 3$). At least 50 cells were randomly selected from each group to score. For (A, B, D, G), the $P$ value was determined by two-tailed Student's $t$ test. NS not significant; *$P < 0.05$; **$P < 0.01$; ***$P < 0.001$. For (E, H, K, L, M, O), the $P$ value was determined by a one-way ANOVA analysis. NS not significant; *$P < 0.05$; **$P < 0.01$; ***$P < 0.001$. Source data are available online for this figure.

aggravated cell vacuolization during proteasome inhibition (Fig. 2A–D). Therefore, we hypothesize that these cells may undergo paraptosis. To identify this hypothesis, we first examined whether the vacuoles are swollen ER using transiently expressed enhanced green fluorescent protein targeted to ER (ER-EGFP). The results showed that ER-EGFP remained in the vacuoles of cells treated with proteasome inhibitors combined with KN-93 (Fig. 2E). Additionally, the mitochondria lost their network structure and clustered around the nucleus in the co-treated cells (Fig. 2E). Consistent with the above results, transmission electron microscopy showed that co-treated cells exhibited ER-derived vacuole expansion. Contrary, MG132 or KN-93 alone treated cells exhibited reticular ER (Fig. 2F). These results indicate that CaMKII inhibition aggravated ER dilation. We found that CHX successfully suppressed the vacuolization of MG132/KN-93-treated cells, suggesting this type of cell damage is dependent on new protein synthesis (Fig. 2G–I), which is the main inducing factor of paraptosis. Previous studies have revealed that N-acetylcysteine (NAC), an oxygen radical scavenger, can inhibit paraptosis (Rozpedek et al, 2016; Chen et al, 2019). Next, NAC was tested to determine whether it could reverse the cell vacuolization and cell damage induced by proteasome inhibitor combining CaMKII inhibitor. The results indicated that NAC successfully suppressed the vacuolization and increased the cell viability, but not Z-VAD-FMK, a pan-caspase inhibitor that can block caspase-dependent cell apoptosis (Fig. 2G–K; Appendix Fig. S1A–C). Similar to apoptosis, the cells under paraptosis can also be labeled by Annexin 5. We stained the cells after treated with MG132 and/or KN-93 and found that co-treatment dramatically increased the percentage of Annexin 5-positive cells, which could be reversed by NAC or CHX (Appendix Fig. S1D,E). Furthermore, the molecular pathways related to paraptosis were detected, revealing that MG132 combined with KN-93 further activated the MAPK signaling pathway, such as the phosphorylation of ERK, MEK and p38, and enhanced the ER stress, such as the upregulation of GRP78 and CHOP, compared to the alone treatment (Fig. 2L). These results indicate that CaMKII inhibition aggravates cell paraptosis during proteasome suppression.

## CaMKII regulates proteasome inhibition-induced proteotoxicity through the HRI/eIF2α signaling axis

Phosphorylation of eIF2α by eIF2α-kinases (EIF2AKs) can shut down the GAP-dependent protein translation in response to cellular stresses, such as ER stress (Rozpedek et al, 2016; Boye et al, 2020). Interestingly, although MG132 combined with KN-93 aggravated the ER stress, we did not observe an increase in eIF2α phosphorylation (Fig. 3A–C). Conversely, KN-93 reversed the eIF2α phosphorylation upregulation induced by proteasome inhibitor (Fig. 3A–C), suggesting that CaMKII activation regulates the eIF2α phosphorylation during proteasome inhibition. Given that eIF2α phosphorylation could inhibit protein synthesis, we hypothesized that under proteasome inhibition, phosphorylated eIF2α might reduce misfolded protein accumulation by weakening the protein translation, thereby resisting proteotoxic crisis. To test this hypothesis, we used an eIF2α inhibitor, ISRIB, to examine the effect of eIF2α on aggregated misfolded protein production and cell fate under proteasome inhibition. The results showed that ISRIB significantly enhanced aggregated ubiquitinated protein accumulation and aggravated the paraptosis of cells treated by MG132 (Fig. EV2A–I), suggesting that eIF2α activation is involved in the cell defense against the proteotoxic crisis.

Since eIF2α is phosphorylation mainly catalyzed by EIF2AKs, consisting of HRI (EIF2AK1), double-stranded RNA (dsRNA)-dependent protein kinase (PKR, EIF2AK2), PKR-like endoplasmic reticulum kinase (PERK, EIF2AK3), and General control nonderepressible-2 (GCN2, EIF2AK4), under different cellular stresses (Boye et al, 2020), we next investigated which kinase is responsible for phosphorylating eIF2α during proteasome suppression. EIF2AKs-specific siRNA was used to knock down EIF2AKs in cells (Fig. EV3A,B), and then its effect was observed on eIF2α phosphorylation level, ubiquitinated protein accumulation, and cell vacuolization. The results showed that HRI knockdown significantly decreased eIF2α phosphorylation induced by the proteasome inhibitor compared to the other three EIF2AKs (Fig. EV3C). Similarly, only HRI knockdown aggregated ubiquitinated protein accumulation in NP-40 insoluble fractions and cell vacuolization during proteasome inhibition (Fig. EV3D–F). These results suggest

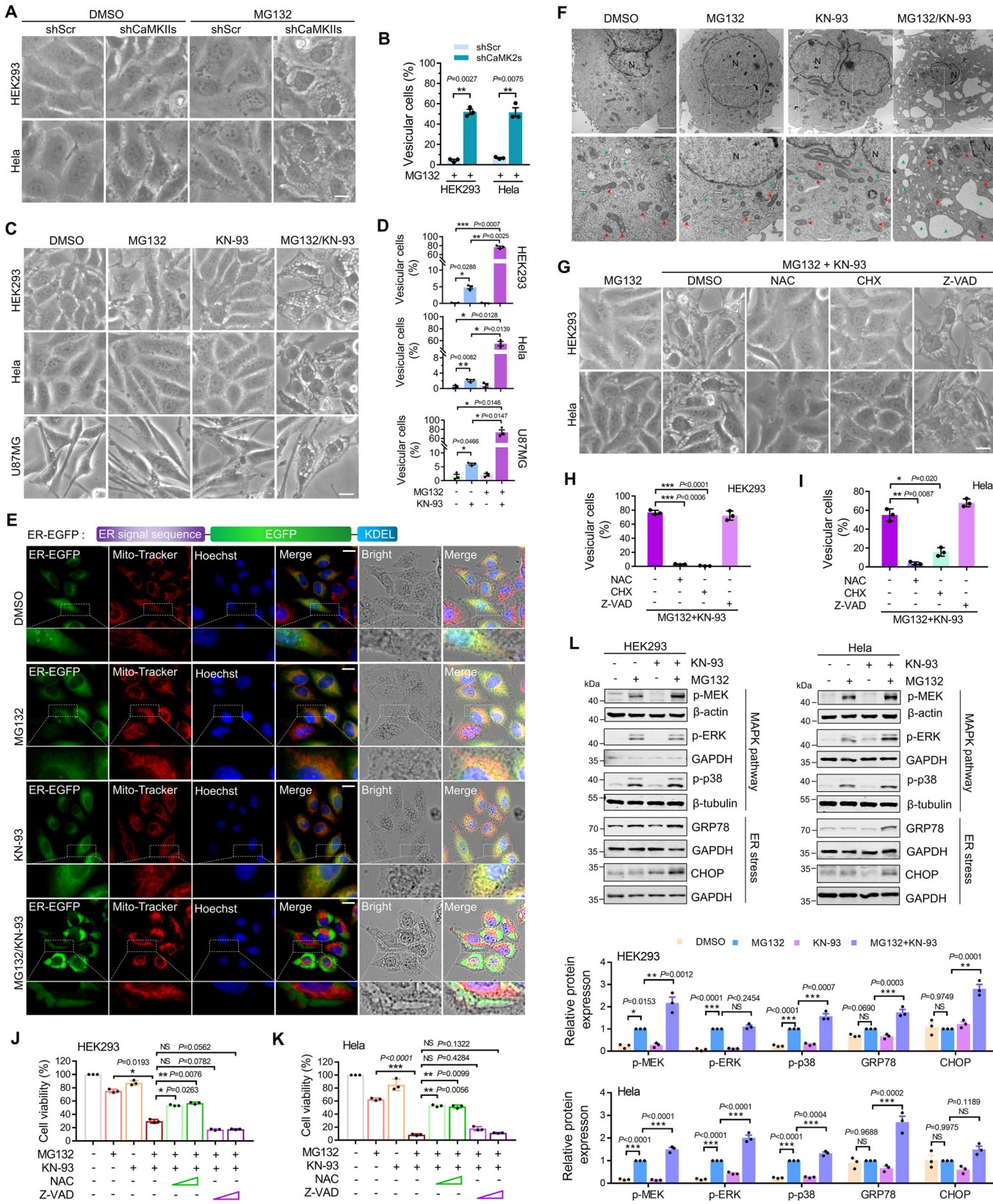

◀ **Figure 2.  Inhibition of CaMKII aggravates proteasome inhibitor-induced paraptosis.**

(A) Cell morphology imaging of HEK293 and HeLa cells transfected with indicated shRNA and treated with MG132 (1 μM) for 24 h. Scale bar: 10 μm. (B) Quantitative analysis of results in (A). Data are mean ± SEM of biological replicates ($n = 3$). At least 50 cells were randomly selected from each group to score. (C) Cell morphology imaging of HEK293, HeLa and U87MG cells treated with indicated drugs (MG132 (1 μM), KN-93 (10 μM)) for 24 h. Scale bar: 10 μm. (D) Quantitative analysis of results in (C). Data are mean ± SEM of biological replicates ($n = 3$). At least 50 cells were randomly selected from each group to score. (E) Representative images of HeLa cells transfected with ER-EGFP and treated with indicated drugs (MG132 (1 μM), KN-93 (10 μM)) for 24 h. Mitochondria and nuclei were stained with Mito-Tracker (red) and Hoechst (blue), respectively. Scale bar: 20 μm. (F) Representative electron micrographs from HeLa cells treated with indicated drugs (MG132 (1 μM), KN-93 (10 μM)) for 24 h. N, nucleus; Red arrows, mitochondria; Green arrows, ER. Scale bar: 2 μm. (G) Cell morphology imaging of HEK293 and HeLa cells treated with indicated drugs (MG132 (1 μM), KN-93 (10 μM), Cycloheximide (CHX, 25 μg/ml), N-Acetylcysteine (NAC, 2 mM), Z-VAD-FMK (Z-VAD, 10 μM)) for 24 h. Scale bar: 10 μm. (H, I) Quantitative analysis of results in (G). Data are mean ± SEM of biological replicates ($n = 3$). At least 50 cells were randomly selected from each group to score. (J, K) Cell viability of HEK293 and HeLa cells after treated with indicated drugs (MG132 (1 μM), KN-93 (10 μM), NAC (2 mM, 4 mM), Z-VAD-FMK (10 μM, 20 μM)) for 48 h. Data are mean ± SEM of biological replicates ($n = 3$). (L) Immunoblotting and quantification of indicated proteins in HEK293 and HeLa cells treated with indicated drugs for 14 h. Data are mean ± SEM of biological replicates ($n = 3$). For (B), the $P$ value was determined by two-tailed Student's $t$ test. **$P < 0.01$. For (D, H–L), the $P$ value was determined by a one-way ANOVA analysis. NS not significant; *$P < 0.05$; **$P < 0.01$; ***$P < 0.001$. Source data are available online for this figure.

that HRI mediates eIF2α phosphorylation in response to proteasome inhibition and suppresses ubiquitinated protein synthesis and proteotoxicity.

Next, we investigated whether CaMKII regulates the cell defense against proteotoxic crisis by activating HRI during proteasome inhibition. Here, an HRI-specific agonist, BTdCPU, was used to reverse the phenotype mediated by the CaMKII inhibitor. The results indicated that BTdCPU could significantly restore eIF2α phosphorylation level downregulated by KN-93 (Fig. 3D). Besides, biochemistry and immunostaining experiments showed BTdCPU could reverse aggregated ubiquitinated protein accumulation induced by CaMKII inhibitor (Fig. 3E–G). In addition, BTdCPU could successfully reverse the paraptosis aggravated by inhibiting CaMKII (Figs. 3H–L and EV3G,H). Interestingly, activating PERK, a specific kinase responsible for phosphorylating eIF2α in response to ER stress through the CCT020312, a PEKR-specific agonist, successfully rescued eIF2α phosphorylation and reduced the accumulation of aggregated ubiquitinated proteins (Appendix Fig. S2A,B), but failed to reverse the cytotoxicity mediated by CaMKII inhibitor (Appendix Fig. S2C,D). This suggests that during proteasome inhibition, PERK and HRI-mediated eIF2α phosphorylation have different regulatory outcomes on cell fate. Taken together, these results indicate that CaMKII suppress aggregated ubiquitinated protein accumulation and paraptosis through the HRI/eIF2α signaling axis in response to proteasome inhibition.

## Proteasome inhibition induces CaMKII phosphorylate of BAG3

BAG3 is involved in eIF2α activation during proteasome inhibition (Carra et al, 2009; Patel et al, 2022). We established the BAG3 knockout cell lines and found that deletion of BAG3 decreased the phosphorylation level of eIF2α (Fig. EV4A,B). Therefore, we next investigated whether CaMKII regulates the HRI-eIF2α signaling pathway through BAG3. First, the potential interaction between CaMKIIs and BAG3 was examined, revealing that proteasome inhibition promoted the binding of endogenous CaMKIIs to BAG3 in HEK293 cells (Fig. 4A). Additionally, FLAG-tagged exogenous CaMKIIβ/δ were overexpressed in HEK293 cells, and then the interaction between these overexpressed CaMKIIs and BAG3 was detected by Co-immunoprecipitation. The results showed that proteasome inhibition also promoted the binding of CaMKIIβ/δ to

BAG3 (Fig. 4B). Furthermore, compared with wild-type CaMKIIs, the binding ability of CaMKIIs (T287A), T287 phosphorylation-deficient mutants, to BAG3 was significantly weakened (Fig. 4C–E). This finding suggests that proteasome inhibition may promote the interaction by activating CaMKIIs. To verify this possibility, constitutively active kinase mutants of CaMKIIβ/δ (D-S), containing only 1-291 amino acids (Hansen et al, 2006; Chao et al, 2010; Saito et al, 2013), were constructed while mutating T287 to mimic phosphorylated Asp (D). Then, the interaction of these mutants was tested with BAG3 under regular proteasome activity, indicating that the ability of activated CaMKIIs to bind BAG3 was significantly enhanced compared to the wild-type CaMKIIs (Fig. 4F,G). These results indicate that proteasome inhibition promotes the binding of CaMKIIs to BAG3 by activating CaMKIIs.

Since activated CaMKII could bind BAG3, BAG3 may be phosphorylated by CaMKII. The effect of proteasome inhibitors, or CaMKII inhibitors, or both on BAG3 phosphorylation was first detected using the Phos-Tag system to test this possibility. Our results showed that proteasome inhibition increased BAG3 phosphorylation level, which could be significantly suppressed by KN-93 (Fig. 4H). This finding suggests that there is CaMKII-dependent BAG3 phosphorylation during proteasome inhibition. To identify the CaMKII phosphorylated site in BAG3, FLAG-tagged BAG3 was overexpressed in HEK293 cells either alone or in combination with constitutively active kinase mutants of CaMKIIs. After treatment with inhibitors, exogenous BAG3 was immunopurified for mass spectrometry analysis (Fig. 4I). In addition to common phosphorylation sites in all samples, BAG3-FLAG co-expressed with activated CaMKIIs contained phosphorylation sites at Ser173/377/386 (Figs. 4J and EV4C–G). Sequence alignment of BAG3 from different species indicated that Ser173/377/386 and their surrounding sequence are well conserved in mammals (Fig. 4K). Subsequently, the non-phosphorylatable mutants of BAG3 were constructed and expressed, in which Ser (S) were replaced by Ala (A) and detected their phosphorylation after proteasome inhibitor treatment. We co-expressed the BAG3 mutants with activated CaMKIIs in BAG3 knockout cells, and detected the phosphorylation level of these mutants. The result showed that simultaneous mutation of S173, S377, and S386 to Ala is required to successfully abolish activated CaMKIIs-mediated BAG3 phosphorylation, rather than a single-site mutation (Figs. 4L and EV4H), suggesting CaMKII phosphorylates BAG3 at S173, S377 and S386.

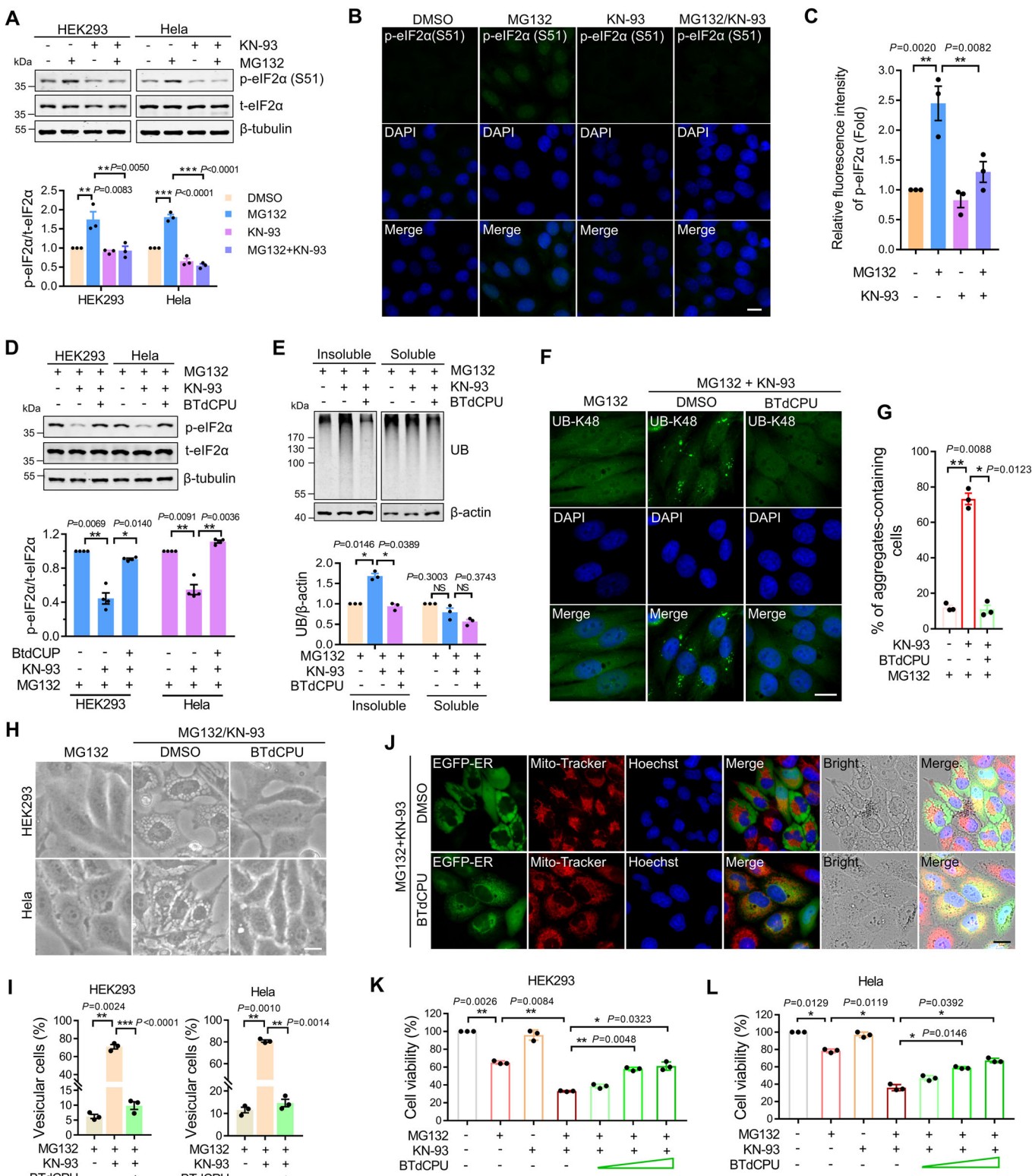

Furthermore, we found that Ser173A/377A/386A (3A) mutation significantly reduced BAG3 phosphorylation levels induced by proteasome inhibitors (Figs. 4M and EV4I). Besides, in vitro phosphorylation assay showed that activated CaMKII could effectively phosphorylate wild-type BAG3 proteins purified from bacteria but not the 3A mutant (Figs. 4N and EV4J). This result confirmed that Ser173A/377A/386A of BAG3 are the major phosphorylation sites of CaMKII.

◄ **Figure 3. CaMKII regulates proteasome inhibition-induced proteotoxicity through the HRI/eIF2α signaling axis.**

(A) Immunoblotting and quantification of p-eIF2α (S51) in HEK293 and HeLa cells treated with indicated drugs (MG132 (1 μM), KN-93 (10 μM)) for 14 h. Data are mean ± SEM of biological replicates (n = 3). (B) Representative p-eIF2α (S51) staining images of HeLa cells treated with indicated drugs (MG132 (1 μM), KN-93 (10 μM)) for 14 h. Scale bar: 20 μm. (C) Immunofluorescence-based measurement and quantification of the expression levels of p-eIF2α shown in (B). Data are mean ± SEM of biological replicates (n = 3). (D) Immunoblotting and quantification of p-eIF2α (S51) in HEK293 and HeLa cells treated with indicated drugs (MG132 (1 μM), KN-93 (10 μM), BTdCPU (2 μM)) for 14 h. Data are mean ± SEM of biological replicates (n = 4). (E) Immunoblotting and quantification of UB in NP-40-soluble and -insoluble fractions of HEK293 cells treated with indicated drugs for 14 h. Data are mean ± SEM of biological replicates (n = 3). (F) Representative UB-K48 staining images of HeLa cells treated indicated drugs for 14 h. Scale bar: 20 μm. (G) Quantitative analysis of results in (F). Data are mean ± SEM of biological replicates (n = 3). At least 50 cells were randomly selected from each group to score. (H) Cell morphology imaging of HEK293 and HeLa cells treated with indicated drugs for 24 h. Scale bar: 10 μm. (I) Quantitative analysis of results in (H). Data are mean ± SEM of biological replicates (n = 3). At least 50 cells were randomly selected from each group to score. (J) Representative images of HeLa cells transfected with ER-EGFP and treated with indicated drugs for 24 h. Mitochondria and nuclei were stained with Mito-Tracker (red) and Hoechst (blue), respectively. Scale bar: 20 μm. (K, L) Cell viability of HEK293 and HeLa cells treated with indicated drugs (MG132 (1 μM), KN-93 (10 μM), BTdCPU (0.5 μM, 1 μM, 2 μM)) for 48 h. Data are mean ± SEM of biological replicates (n = 3). For (A, C, D, E, G, I, K, L), the P value was determined by a one-way ANOVA analysis. NS not significant; *P < 0.05; **P < 0.01; ***P < 0.001. Source data are available online for this figure.

## CaMKII activates HRI-eIF2α through phosphorylating BAG3 during proteasome inhibition

Since binding to molecular chaperons is a crucial way to activate HRI, we hypothesize that phosphorylated BAG3 may mediate HRI activation in response to proteasome inhibition. Subsequently, the interaction between BAG3 and HRI was examined, and the co-immunoprecipitation showed that proteasome inhibition induced the binding of BAG3 to HRI, while CaMKII inhibitor KN-93 suppressed that (Fig. 5A). This result aligns with the immunostaining results, where KN-93 eliminated the colocalization of BAG3 and HRI induced by proteasome inhibitors (Fig. 5B). In addition, overexpression of wild-type CaMKIIβ, but not kinase death mutant (CaMKIIβ (K43M)), in HEK293 cells promoted BAG3 binding to HRI (Fig. 5C). Next, the effect of BAG3 phosphorylation on the interaction was investigated, revealing that the re-expression of BAG3 3A mutant in BAG3 knockout cells significantly weakened the interaction between BAG3 and HRI, compared to the re-expression of wild-type BAG3 or Ser173D/377D/386D (3D) mutant, which mutated Ser to phospho-mimetic Asp (D) residuals (Fig. 5D). This result suggests that the binding of BAG3 to HRI induced by proteasome inhibitor depends on Ser173/377/386 phosphorylation.

Previous study reported that the binding of BAG3 to HSPs was involved in the HRI-associated eIF2α phosphorylation (Patel et al, 2022). We then detected the effect of CaMKII activation on the complex formation of BAG3-HSPs, and found that inhibiting CaMKII did not affect the interaction of BAG3 and HSP70 (HSPA1A) and HSPB8 (Appendix Fig. S3A–C), two important BAG3-associated HSPs. Although the specific molecular mechanism of HRI activation is still unclear, it has been reported that the oligomerization and autophosphorylation of HRI can promote its activation (Rafie-Kolpin et al, 2003; Shao et al, 2003; Bauer et al, 2001; Ricketts et al, 2022). Therefore, the effect of CaMKII inhibition and BAG3 phosphorylation on HRI phosphorylation level was examined. The results showed that inhibiting CaMKII significantly decreased HRI phosphorylation levels induced by proteasome inhibitor (Fig. 5E). Furthermore, the wild-type BAG3 and BAG3 3D mutants could enhance HRI phosphorylation during proteasome inhibition compared to the BAG3 3A mutant (Fig. 5F). Besides, in BAG3 knockout cells, re-expression of BAG3 3D mutant can successfully rescue proteasome inhibition-mediated eIF2α phosphorylation compared to BAG3 3A mutant (Fig. 5G,H). Consistent with the failure of eIF2α phosphorylation, the BAG3 3A

mutant aggravated the protein synthesis when proteasome was inhibited compared to wild-type BAG3 and 3D mutant (Fig. 5I). Collectively, these results suggest that CaMKIIs activate the HRI-eIF2α signaling axis through phosphorylating BAG3 under proteasome inhibition, thereby suppressing the production of aggregated ubiquitinated proteins.

## CaMKII regulates proteasome inhibition-induced proteotoxicity through phosphorylating BAG3

We found that knockout of BAG3 significantly aggravated the cell vacuolization induced by proteasome inhibition (Appendix Fig. S4A–D). Next, we examined the effect of BAG3 phosphorylation on proteasome inhibition-induced cellular proteotoxicity. The results showed that re-expression of BAG3 3A enhanced ubiquitinated misfolded protein accumulation in NP-40 insoluble fractions during proteasome inhibition compared to the wild-type BAG3 (Fig. 6A; Appendix Fig. S4E). Immunostaining experiments also indicated that BAG3 3A enhanced aggregated ubiquitinated protein accumulation (Fig. 6B,C). Besides, the 3A mutation aggravated the vacuolization and cell damage induced by proteasome inhibitors (Fig. 6D–F). Next, we tested whether the phosphomimetic mutant could reverse aggregated ubiquitinated protein accumulation and paraptosis caused by CaMKII inhibition. The results revealed that BAG3 3D significantly reduced aggregated ubiquitinated protein accumulation and paraptosis-associated cell damage during proteasome and CaMKII inhibition (Fig. 6F–K). In addition, we also observed that only the BAG3 3D mutation could successfully rescue the intensification of aggregated ubiquitinated protein accumulation caused by KN-93, rather than the single-point phosphomimetic mutants (Appendix Fig. S4F), which further confirms the phosphorylation behavior of CaMKII on the BAG3 S173/S377/S386 sites. Taken together, our study revealed that CaMKII activates the HRI-eIF2α signaling axis through phosphorylating BAG3 under proteasome inhibition, thereby suppressing the production of aggregated ubiquitinated proteins and their cytotoxicity (Fig. 6L).

## Inhibition of CaMKII enhances the anti-tumor activity of proteasome inhibitor

Given that CaMKII alleviate the proteotoxicity induced by proteasome inhibition, concurrently inhibiting CaMKII could

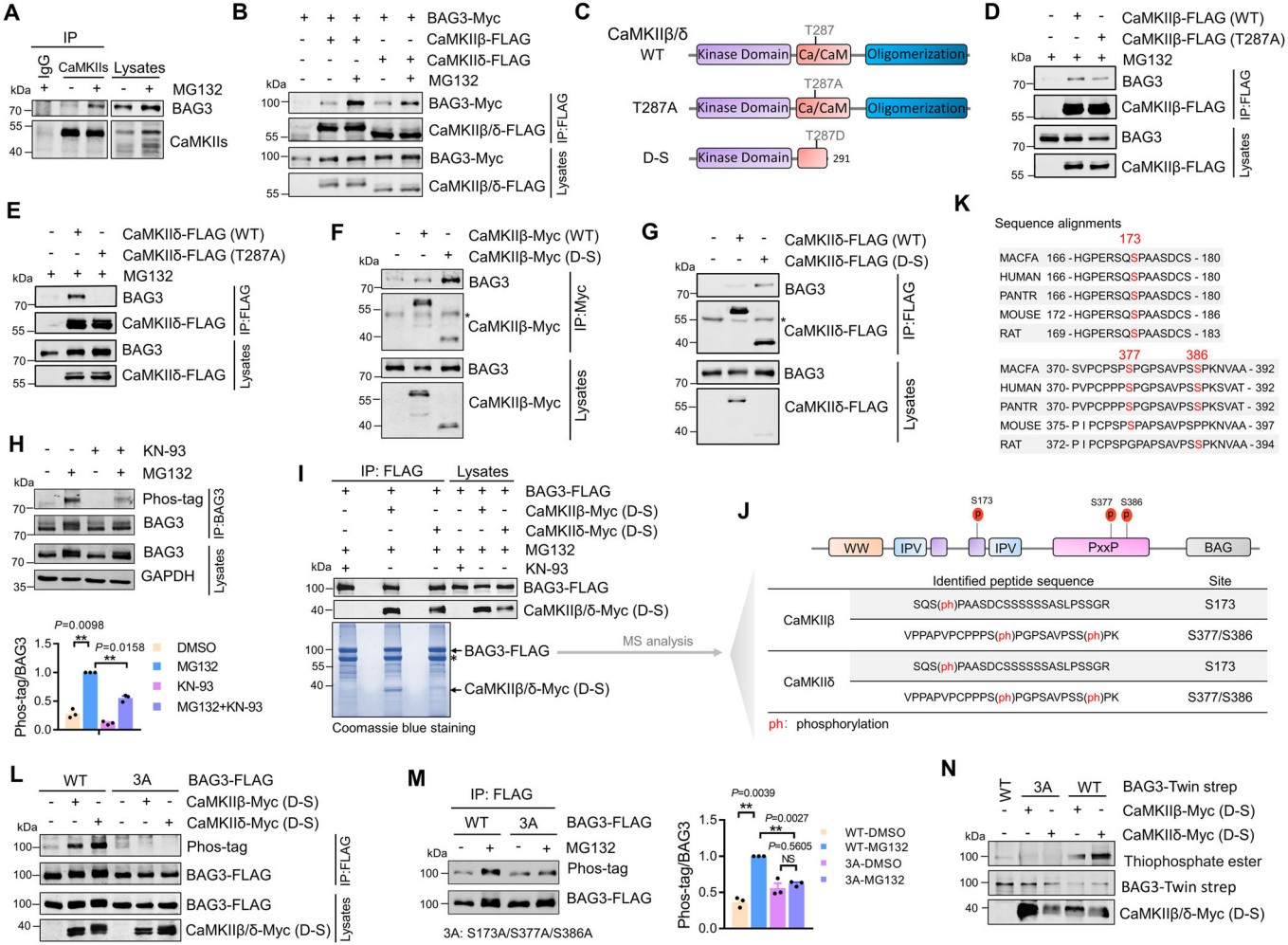

**Figure 4. Proteasome inhibition induces CaMKIIs phosphorylate of BAG3.**

(A) Interaction between endogenous CaMKIIs and BAG3 in HEK293 cells. (B) Interaction between exogenous CaMKIIs and BAG3 in HEK293 cells. HEK293 cells transfected with indicated plasmids were treated with MG132 or not, immunoprecipitation was done using indicated antibody. (C) Illustration of CaMKIIβ/δ domain organization and respective mutants. (D–G) Interaction between CaMKII mutants and BAG3 in HEK293 cells. HEK293 cells transfected with indicated plasmids were treated with MG132 (D, E) or not (F, G), immunoprecipitation was done using indicated antibody. (H) Phosphorylation level of BAG3 in HEK293 cells treated with indicated drugs. (I, J) Identification of CaMKIIs phosphorylation site in BAG3 by tandem MS/MS spectra. After transfected, HEK293 cells were treated with indicated drugs. The BAG3-FLAG proteins were immunopurified and subjected to mass spectrometry analysis. Asterisks indicate unspecific bands (K) Alignment of BAG3 protein sequences across multiple species. (L, M) Phosphorylation level of re-expressed BAG3 (WT or mutants) in BAG3⁻/⁻ HEK293 cells treated with MG132 (M) or not (L). 3 A, S173A/S377A/S386A. (N) In vitro kinase assays containing immunopurified CaMKIIβ-Myc (D-S), CaMKIIδ-Myc (D-S), and 1 μg wild-type or mutated BAG3-Twin Strep, as indicated were incubated with ATP-γ-S. (L, N) are representative results of $n = 2$ independent experiments with similar tendency, and I is the result of $n = 1$ experiment. For (H) and (M), data are mean ± SEM of biological replicates ($n = 3$), and the $P$ value was determined by a one-way ANOVA analysis and Student's $t$ test (two-sided), respectively. NS not significant; **$P < 0.01$. Source data are available online for this figure.

enhance the anti-tumor efficacy of proteasome inhibitors. Next, we investigated whether targeting CaMKII could enhance the anti-tumor efficacy of proteasome inhibitor bortezomib. First, the effect of the combination of Bortezomib and KN-93 on tumor viability was examined. The results revealed that combined with KN-93 can enhance the sensitivity of HeLa, MDA-MB-231, U87MG, LN229, U251, and GL261 cells to Bortezomib (Fig. 7A–F). Besides, the combination aggravated aggregated ubiquitinated protein accumulation in tumor cells (Figs. 7G and EV5A). The results of cell morphological observation showed that KN-93 significantly aggravated tumor cell vacuolization induced by bortezomib, which could be blocked by protein translation inhibitor CHX

(Fig. EV5B–G). Similarly, the cell damage aggravated by KN-93 also could be reversed by NAC compared to caspase inhibitor Z-VAD-FMK (Fig. 7H–J). These results indicate that the combination of KN-93 can aggravate bortezomib-induced proteotoxicity of tumor cells. Moreover, combined with KN-93, decreased eIF2α phosphorylation level induced by bortezomib (Fig. EV5H), and the aggravated cell damage could be reversed by HRI agonist BTdCPU (Fig. 7K–M). This suggests that KN-93 enhanced the anti-tumor activity of bortezomib by inhibiting HRI/eIF2α signaling pathway. To further determine the effect of combined targeting of CaMKII and proteasome on tumor therapy, MDA-MB-231 xenografts were established and treated with Bortezomib and KN-93, alone or in

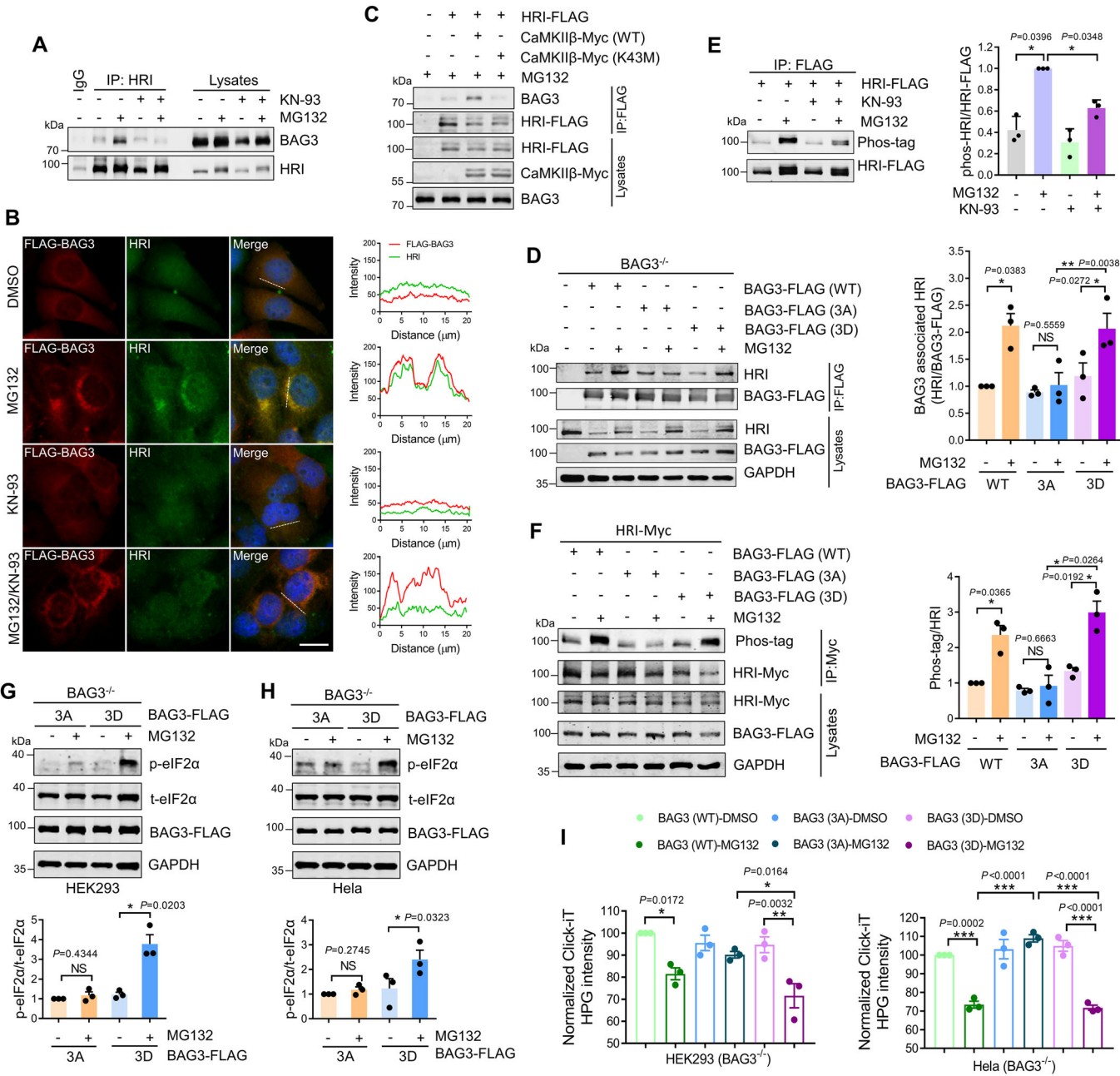

**Figure 5. CaMKII activates HRI through phosphorylating BAG3.**

(A) Interaction between endogenous BAG3 and HRI in HEK293 cells. (B) Representative HRI and FLAG staining images of HeLa cells that were transfected with BAG3-FLAG and treated with indicated drugs. Scale bar: 10 μm. The fluorescence intensity line is tracing corresponding to a white line. (C) The effect of kinase dead mutant of CaMKIIβ on the interaction between BAG3 and HRI. After transfected, HEK293 cells were treated with MG132, and immunoprecipitated using anti-FLAG antibody. (D) Interaction between BAG3 and HRI in BAG3$^{-/-}$ HEK293 cells that were transiently re-expressed wild-type or mutated BAG3 and then treated with MG132. Immunoprecipitation was performed using anti-FLAG antibody. Quantitative analysis of results is showed as the ratio of co-immunoprecipitated HRI to BAG3-FLAG. Data are mean ± SEM of biological replicates ($n = 3$). (E) Phosphorylation level of HRI in HEK293 cells transfected with HRI-FLAG and treated with indicated drugs. Data are mean ± SEM of biological replicates ($n = 3$). (F) Phosphorylation level of HRI in BAG3$^{-/-}$ HEK293 cells that were co-transfected with Myc-tagged HRI and wild-type or mutated BAG3 and then treated with MG132 or not. Data are mean ± SEM of biological replicates ($n = 3$). (G–H) Immunoblotting and quantification of p-eIF2α (S51) in BAG3$^{-/-}$ HEK293 and BAG3$^{-/-}$ HeLa cells that were transiently re-expressed mutated BAG3 and then treated with MG132. Data are mean ± SEM of biological replicates ($n = 3$). (I) Protein synthesis was detected using a Click-iT HPG method in BAG3$^{-/-}$ HEK293 and BAG3$^{-/-}$ HeLa cells transiently re-expressed wild-type or mutated BAG3 and then treated with MG132. Data are mean ± SEM of biological replicates ($n = 3$). For (E, I), the P value was determined by a one-way ANOVA analysis. NS not significant; *$P < 0.05$; **$P < 0.01$; ***$P < 0.001$. For (D, F–H), the P value was determined by Student's t test (two-sided). NS not significant; *$P < 0.05$. Source data are available online for this figure.

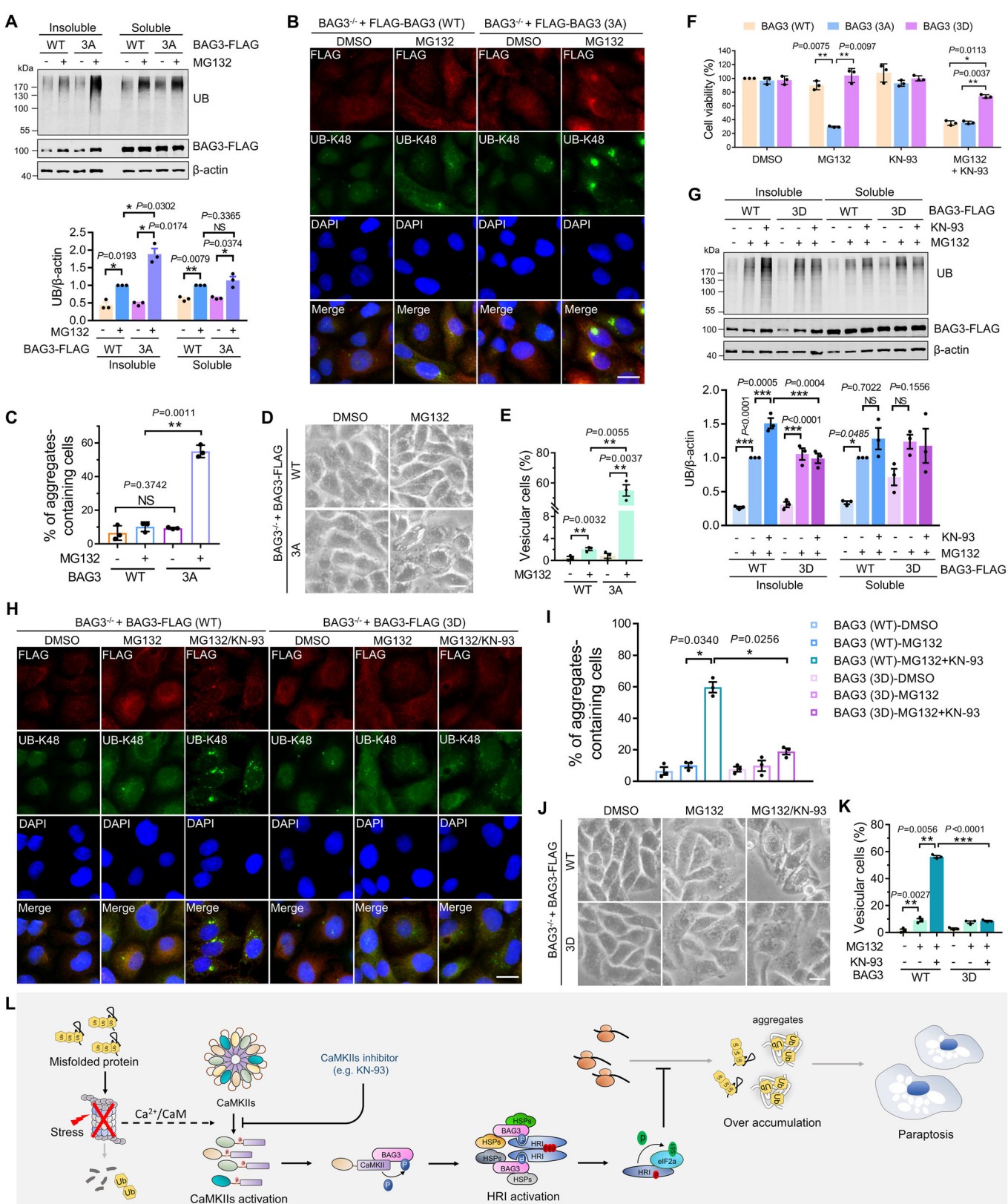

 **Figure 6. CaMKII regulates proteasome inhibition-induced proteotoxicity through phosphorylating BAG3.**

(A) Immunoblotting and quantification of UB in NP-40-soluble and -insoluble fractions of BAG3$^{-/-}$ HEK293 cells that were transiently re-expressed BAG3 (WT) or BAG3 (3A) and then treated with indicated drugs. Data are mean ± SEM of biological replicates ($n = 3$). (B) Representative UB-K48 and FLAG staining images of BAG3$^{-/-}$ HeLa cells that were transiently re-expressed BAG3 (WT) or BAG3 (3A) and then treated with indicated drugs. Scale bar: 20 μm. (C) Quantitative analysis of results in (B). Data are mean ± SEM of biological replicates ($n = 3$). At least 50 cells were randomly selected from each group to score. (D) Cell morphology imaging of BAG3$^{-/-}$ HeLa cells were transiently re-expressed BAG3 (WT) or BAG3 (3A) and then treated with indicated drugs for 24 h. Scale bar: 10 μm. (E) Quantitative analysis of results in (D). Data are mean ± SEM of biological replicates ($n = 3$). At least 50 cells were randomly selected from each group to score. (F) Cell viability of BAG3$^{-/-}$ HEK293 cells that were transiently re-expressed BAG3 (WT), BAG3 (3A) or BAG3 (3D) and then treated with indicated drugs for 48 h. (G) Immunoblotting and quantification of UB in NP-40-soluble and -insoluble fractions of BAG3$^{-/-}$ HEK293 cells that were transiently re-expressed BAG3 (WT) or BAG3 (3D) and then treated with indicated drugs. Data are mean ± SEM of biological replicates ($n = 3$). (H) Representative UB-K48 and FLAG staining images of BAG3$^{-/-}$ HeLa cells that were transiently re-expressed BAG3 (WT) or BAG3 (3D) and then treated with indicated drugs. Scale bar: 20 μm. (I) Quantitative analysis of results in (H). Data are mean ± SEM of biological replicates ($n = 3$). At least 50 cells were randomly selected from each group to score. (J) Cell morphology imaging of BAG3$^{-/-}$ HeLa cells were transiently re-expressed BAG3 (WT) or BAG3 (3D) and then treated with indicated drugs for 24 h. Scale bar: 10 μm. (K) Quantitative analysis of results in (J). Data are mean ± SEM of biological replicates ($n = 3$). At least 50 cells were randomly selected from each group to score. For (A, C, E), the P value was determined by Student's t test (two-sided). NS not significant; *$P < 0.05$; **$P < 0.01$. For (F, G, I, K), the P value was determined by a one-way ANOVA analysis. NS not significant; *$P < 0.05$; **$P < 0.01$; ***$P < 0.001$. (L) Model depicting proteasome inhibition-induced activation of CaMKII regulates the proteotoxicity by phosphorylating BAG3 and activating HRI-eIF2α pathway. Source data are available online for this figure.

combination. The results indicated that dual Bortezomib and KN-93 treatment significantly enhanced the anti-tumor activity compared to Bortezomib or KN-93 alone (Fig. 7N–P). However, bortezomib-induced toxic effects regarding weight loss (Fig. EV5I). This implies that further in-depth studies are warranted before clinical application. Overall, these results suggest a new therapeutic intervention strategy for proteasome inhibitor application in tumor treatment.

## Discussion

Misfolded protein quality control is critical for cells to defend against proteotoxic crises in response to proteasome impairment, which contributes to developing and treating several diseases, including neurodegenerative diseases, cancer, and cardiomyopathy (Moreno-Gonzalez et al, 2011). Herein, proteasome inhibition activated CaMKII and promoted its phosphorylation of BAG3 at Ser173/377/386 sites. Moreover, phosphorylated BAG3 could inhibit the production of aggregated misfolded proteins by activating the HRI/eIF2α signaling pathway, thereby attenuating cytotoxic damage (Fig. 6L).

Aggregation is a crucial behavioral change in ubiquitinated misfolded proteins when proteasome activity is decreased or inhibited (Goldberg, 2003; Ciechanover, 2005). Although the molecular mechanism behind ubiquitinated protein aggregation is unclear, BAG3 is considered involved in this process. Disrupting BAG3/HSP70 complex formation can suppress the aggregation and autophagic sequestration of ubiquitinated misfolded proteins during proteasome inhibition (Minoia et al, 2014; Sha et al, 2009; Gamerdinger et al, 2011). The accumulation of ubiquitinated proteins in cells exacerbated by BAG3 depletion during proteasome inhibition is caused by insufficient aggrephagy (Minoia et al, 2014; Gamerdinger et al, 2011). However, our previous studies have shown that aggrephagy minimally affects the quality control of ubiquitinated proteins in culture cells under proteasome inhibition (Zhang et al, 2022a; Zhang et al, 2023). Furthermore, BAG3 regulates protein synthesis-associated quality control of ubiquitinated misfolded proteins in response to proteasome impairment (Carra et al, 2009; Patel et al, 2022). Additionally, Prosser et al, found that aggresome formation induced by proteasome inhibitors

depend on protein synthesis (Prosser et al, 2022). These results suggest that BAG3 may regulate misfolded protein aggregation by controlling the protein synthesis rate during proteasome inhibition. This study found that BAG3 could inhibit ubiquitinated misfolded protein synthesis by regulating the activation of the HRI-eIF2α signaling axis. These results confirm that misfolded protein synthesis is a critical stage in the quality control of ubiquitinated protein aggregation.

The primary way for BAG3 to facilitate proteasome inhibition-induced ubiquitinated protein aggregation is by forming a complex with HSPs (Stürner et al, 2017; Minoia et al, 2014; Sha et al, 2009; Gamerdinger et al, 2011; Zhang et al, 2011). Herein, we confirmed that BAG3 could decrease aggregated misfolded protein accumulation by activating the HRI/eIF2α signaling axis. Moreover, we found that CaMKII regulated ubiquitinated misfolded protein production through phosphorylating BAG3. However, CaMKII did not affect the association of BAG3 and HSPs, suggesting that reducing this accumulation by CaMKII depends on misfolded protein synthesis upregulation. These results indicated that as an effector of proteasome impairment, BAG3 regulates ubiquitinated misfolded protein aggregation in two opposite manners. The aggregation of misfolded proteins mediated by the BAG3/HSPs complex protects the cells from cytotoxic protein species. Moreover, suppressing misfolded protein synthesis regulated by the BAG3-HRI-eIF2α signal pathway reduces misfolded protein production in cells. Their joint actions attenuate the toxicity of misfolded proteins to cells during proteasome inhibition.

Regarding cellular stresses, eIF2α is a vital protein translation regulator. Although the phosphorylation level of eIF2α increased due to proteasome inhibition, the specific molecular mechanism is unclear. Yerlikaya et al, reported that this upregulation of eIF2α phosphorylation induced by proteasome inhibitors in mouse embryonic cells depended on HRI (Yerlikaya et al, 2008). Furthermore, Alvarez et al, recently showed that proteasome inhibitors can reduce protein synthesis in mouse brain tissue by accelerating HRI-mediated eIF2α phosphorylation (Alvarez-Castelao et al, 2020). Herein, by knocking down four different EIF2AKs, we also revealed that proteasome inhibition-induced eIF2α phosphorylation depends on HRI activation, consistent with the previous studies. PERK, an essential eIF2α kinase, is usually activated during ER stress and then transduces the cytosolic signal

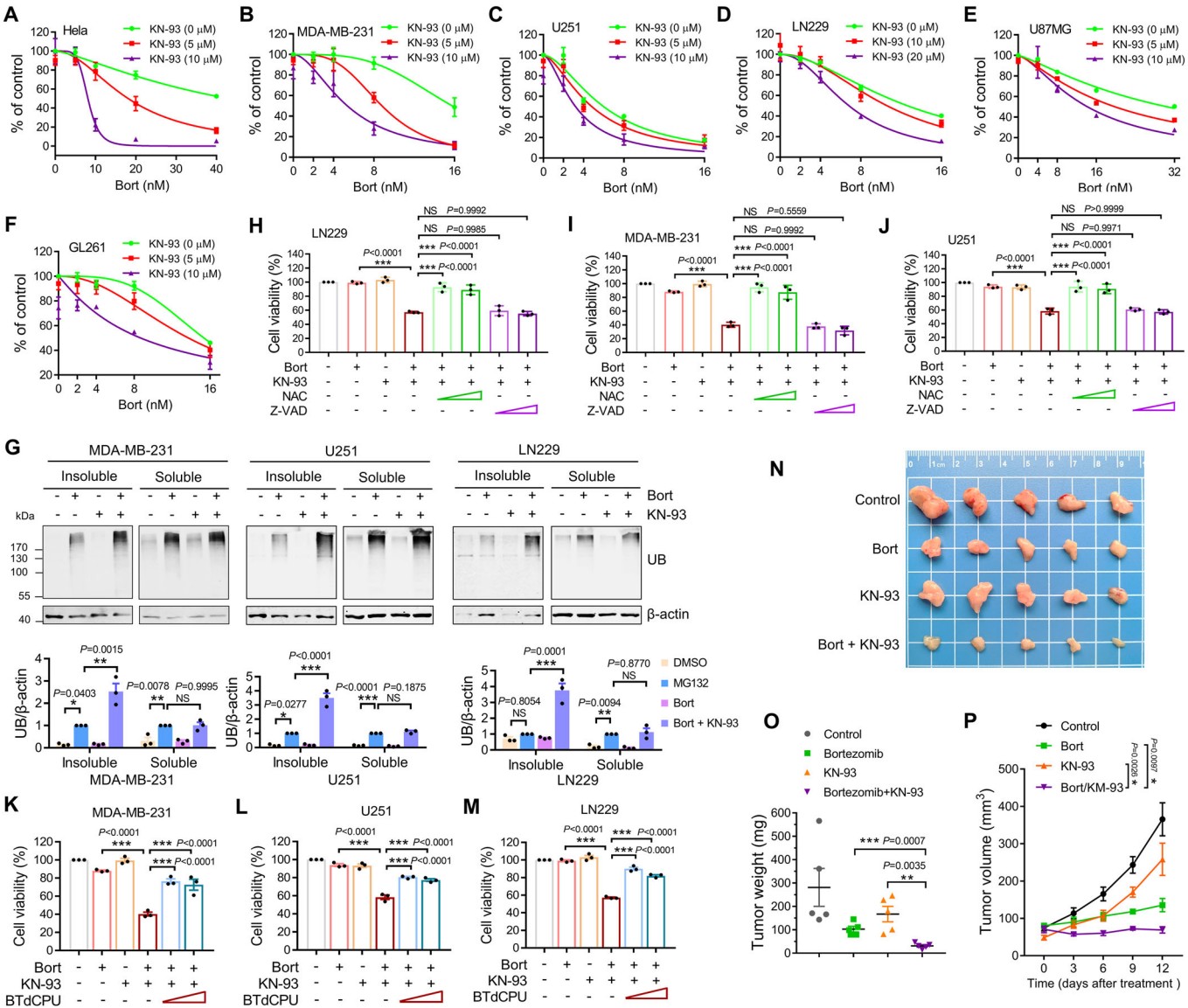

**Figure 7. Inhibition of CaMKII enhances the anti-tumor activity of proteasome inhibitor.**

(A–F) Dose-response curves of tumor cells to Bortezomib (Bort) in combination with KN-93. Data were from three independent experiments ($n = 3$); error bar: SEM. (G) Immunoblotting and quantification of UB in NP-40-soluble and -insoluble fractions of tumor cells treated with indicated drugs (Bort (10 nM), KN-93 (10 μM)) for 14 h. Data are mean ± SEM of biological replicates ($n = 3$). The $P$ value was determined by a one-way ANOVA analysis. NS, not significant; *$P < 0.05$; **$P < 0.01$; ***$P < 0.001$. (H–J) Cell viability of tumor cells treated with indicated drugs (Bort (10 nM), KN-93 (5 μM), NAC (1 mM, 2 mM), Z-VAD-FMK (5 μM, 10 μM)) for 48 h. Data are mean ± SEM of biological replicates ($n = 3$). The $P$ value was determined by a one-way ANOVA analysis. NS, not significant; ***$P < 0.001$. (K–M) Cell viability of tumor cells treated with indicated drugs ((Bort (10 nM), KN-93 (5 μM), BTdCPU (0.25 μM, 0.5 μM)) for 48 h. Data are mean ± SEM of biological replicates ($n = 3$). The $P$ value was determined by a one-way ANOVA analysis. NS not significant; ***$P < 0.001$. (N–P) Anticancer effect of the combination of Bort and KN-93 in MDA-MB-231 cell subcutaneous tumor model. (N) Excised tumors after 12-day treatment. (O) Tumor weights of excised tumors. Data are mean ± SEM of five mice. The $P$ value was determined by Student's $t$ test (two-sided). *$P < 0.05$; **$P < 0.01$. (N) Tumor volume of each group. Data are mean ± SEM of five mice. The $P$ value was determined by Student's $t$ test (two-sided) for comparison at day 12. *$P < 0.01$, **$P < 0.001$. Source data are available online for this figure.

between ER stress and gene expression (Fribley et al, 2006; Zhang et al, 2010; Ribeiro et al, 2019). Interestingly, although proteasome inhibition can induce ER stress and activate PERK, PERK depletion neither affects eIF2α phosphorylation level nor aggregated ubiquitinated protein accumulation, suggesting that eIF2α has a strict choice for its kinase during proteasome inhibition. In addition, in this study, we found activating PERK can reverse eIF2α phosphorylation inhibition and the accumulation of

aggregated ubiquitinated proteins caused by MG132/KN-93 treatment, but not the cell damage (Appendix Fig. S2). This suggests that during proteasome inhibition, PERK and HRI-mediated eIF2α phosphorylation have different regulatory outcomes on cell fate. One possible mechanism is that over-activated PERK may trigger apoptosis.

Herein, we revealed that the binding of BAG3 to HRI is essential for activating the HRI/eIF2α signaling axis. Although the molecular

structure basis of HRI activation is still unclear, binding to activators, such as molecular chaperones, is an essential model for activating HRI (Uma et al, 1999; Berwal et al, 2018; Kulkarni et al, 2010). Additionally, HRI oligomerization could enhance its autophosphorylation, thereby activating its protein kinase activity (Donnelly et al, 2013). Serena Carra et al, reported that BAG3 overexpression in HEK293 cells enhanced eIF2α phosphorylation and reduced misfolded protein production and found that this function of BAG3 depended on its PxxP motif (Carra et al, 2009). Here, we showed that phosphorylated BAG3 could bind to HRI and increase HRI phosphorylation levels. Combined with the oligomerization behavior of BAG3 under proteasome inhibition, we speculate that the HRI activation model may be that phosphorylated BAG3, at S377/386 in the PxxP motif, recruits and promotes the oligomerization and autophosphorylation of HRI. It should be noted that we found phosphorylated BAG3 activates HRI only under proteasome inhibition. Under normal conditions, phosphorylated BAG3 does not promote HRI activation. This suggests that other signals or molecules activated by proteasome inhibition might also be required for HRI activation. Liquid phase separation (LLPS)- or liquid condensates-dependent kinase activation has recently been widely reported, providing a new molecular mechanism model for protein kinase activation and substrate phosphorylation in the cytoplasm, which contributes to particular disease development, such as LLPS accelerated Tau protein phosphorylation in Alzheimer's disease, and liquid condensates drove kinase fusion activation in cancer cells (López-Palacios et al, 2023; Sang et al, 2022; Tulpule et al, 2021; Sampson et al, 2021). Further studies are needed to determine whether HRI activation by BAG3 also depends on LLPS or liquid condensates during proteasome inhibition.

Paraptosis is a programmed cell death type with an unclear mechanism basis. However, paraptosis was induced by excessive accumulation of ubiquitinated proteins and intracellular oxidative stress. Multiple natural products or synthetic compounds can induce tumor cell paraptosis by inhibiting proteasome and enhancing tumor cell sensitivity to proteasome inhibitors (Lee et al, 2016). This suggests the presence of a particular mechanism suppressing paraptosis during proteasome impairment. Herein, CaMKIIs inhibition induced the dilation of the ER and aggravated paraptosis. Moreover, CaMKII inhibited paraptosis during proteasome impairment by activating BAG3/HRI/eIF2α signal axis-mediated the suppression of protein synthesis and misfolded protein aggregation. Herein, we provide a new molecular mechanism for cell defense against proteotoxic crisis caused by proteasome insufficiency, which may advance our understanding of disease mechanism and therapy.

While inhibiting the accumulation of misfolded protein can slow the progression of certain diseases, like neurodegenerative diseases (Mathieu et al, 2020), it can have a different effect on tumor cells. Reducing the accumulation of misfolded proteins can actually strengthen the resistance of tumor cells to proteasome inhibitors, diminishing their therapeutic effect (Tundo et al, 2020). This is a key factor limiting the use of proteasome inhibitors in solid tumors. Consequently, a potential strategy to overcome drug resistance might be to increase the build-up of misfolded proteins in tumor cells caused by proteasome inhibitors, and thus intensify the resulting proteotoxicity. In this study, based on these molecular mechanisms, we revealed that the combined targeting of CaMKII

significantly enhanced the efficacy of proteasome inhibitors in tumor therapy.

In summary, we revealed the molecular mechanism whereby CaMKII suppresses the proteotoxicity during proteasome inhibition. Briefly, CaMKII promotes the binding of BAG3 to HRI by phosphorylating BAG3, which activates the HRI/eIF2α signaling pathway and inhibits the synthesis of aggregated ubiquitinated proteins, thereby suppressing cell paraptosis. Combining CaMKII inhibitor and proteasome inhibitor aggravates tumor cell death effectively. Therefore, our results provide a novel therapeutic intervention strategy for proteasome inhibitors-mediated tumor suppression.

# Methods

### Reagents and tools table

| Reagent/resource | Reference or source | Identifier or catalog number |
|---|---|---|
| **Experimental models: cell lines** | | |
| HEK293 | ATCC | N/A |
| HeLa | ATCC | N/A |
| MDA-MB-231 | ATCC | N/A |
| U87MG | OriCell | Cat#H8-0501 |
| U251 | OriCell | Cat#H8-0401 |
| LN229 | OriCell | Cat#H8-0301 |
| GL261 | OriCell | Cat#M8-0301 |
| **Recombinant DNA** | | |
| pcDNA3.1(+)-BAG3-4FLAG | Zhang et al, 2022b | N/A |
| pcDNA3.1(+)-BAG3-3Myc | Zhang et al, 2022b | N/A |
| pcDNA3.1(+)-BAG3-EGFP | This study | N/A |
| pcDNA3.1(+)-CaMKIIβ-4FLAG | This study | N/A |
| pcDNA3.1(+)-CaMKIIβ-3Myc | This study | N/A |
| pcDNA3.1(+)-CaMKIIδ-4FLAG | This study | N/A |
| pcDNA3.1(+)-CaMKIIδ-3Myc | This study | N/A |
| pcDNA3.1(+)-HRI-4FLAG | This study | N/A |
| pcDNA3.1(+)-HRI-3Myc | This study | N/A |
| pcDNA3.1(+)-4FLAG-HSPB8 | This study | N/A |
| pcDNA3.1(+)-BAG3 (S173A)-4FLAG | This study | N/A |
| pcDNA3.1(+)-BAG3 (S377A)-4FLAG | This study | N/A |
| pcDNA3.1(+)-BAG3 (S386A)-4FLAG | This study | N/A |
| pcDNA3.1(+)-BAG3 (S173/S377/ S386A)-4FLAG | This study | N/A |
| pcDNA3.1(+)-BAG3 (S173D)-4FLAG | This study | N/A |
| pcDNA3.1(+)-BAG3 (S377D)-4FLAG | This study | N/A |
| pcDNA3.1(+)-BAG3 (S386D)-4FLAG | This study | N/A |
| pcDNA3.1(+)-BAG3 (S173/S377/ S386D)-4FLAG | This study | N/A |
| pcDNA3.1(+)-CaMKIIβ (T287A)- 4FLAG | This study | N/A |

| Reagent/resource | Reference or source | Identifier or catalog number |
|---|---|---|
| pcDNA3.1(+)-CaMKIIδ (T287A)-4FLAG | This study | N/A |
| pcDNA3.1(+)-CaMKIIβ (D-S)-3Myc | This study | N/A |
| pcDNA3.1(+)-CaMKIIδ (D-S)-3Myc | This study | N/A |
| pcDNA3.1(+)-CaMKIIδ (D-S)-FLAG | This study | N/A |
| pcDNA3.1(+)-CaMKIIβ (K43M)-3Myc | This study | N/A |
| pcDNA3.1(+)-ER-EGFP | This study | N/A |
| pET28a-BAG3 (WT)-Twin strep | This study | N/A |
| pET28a-BAG3 (S173/S377/S386A)-Twin strep | This study | N/A |
| pLKO.1-shCAMK2A | This study | N/A |
| pLKO.1-shCAMK2B | This study | N/A |
| pLKO.1-shCAMK2G 1# | This study | N/A |
| pLKO.1-shCAMK2G 2# | This study | N/A |
| pLKO.1-shCAMK2D 1# | This study | N/A |
| pLKO.1-shCAMK2D 2# | This study | N/A |
| pLKO.1-shScr | Addgene | 17920 |
| **Antibodies** | | |
| Rabbit anti-UB-K48 | Millipore | Cat#05-1307 |
| Rabbit anti-UB-K63 | Abcam | Cat#ab179434 |
| Rabbit anti-ubiquitin | Abcam | Cat#ab134953 |
| Rabbit anti-pCaMKII (T286) | Cell Signaling | Cat#12716 |
| Rabbit anti-pCaMKII (T287) | Abcam | Cat#ab182647 |
| Rabbit anti-CaMKII | Cell Signaling | Cat#4436 |
| Mouse anti-CaMKII | Santa Cruz | Cat#sc-5306 |
| Rabbit anti-FLAG tag | Proteintech | Cat#80010-1-RR |
| Mouse anti-FLAG tag | Prospec | Cat#ANT-146-b |
| Mouse anti-Myc Tag | Biolegend | Cat#MMS-150R |
| Mouse anti-GAPDH | Zen Bioscience | Cat#200306 |
| Mouse anti-β-actin | Zen Bioscience | Cat#200068-6D7 |
| Mouse anti-GFP | Proteintech | Cat#66002-1-Ig |
| Rabbit anti-BAG3 | Proteintech | Cat#10599-1-AP |
| Rabbit anti-HSPA1A | Proteintech | Cat#10995-1-AP |
| Rabbit anti-HRI | Proteintech | Cat#20499-1-AP |
| Rabbit anti-PERK | Abcam | Cat#ab229912 |
| Rabbit anti-GCN2 | Abcam | Cat#ab134053 |
| Rabbit anti-PKR | Abcam | Cat#ab32506 |
| Rabbit anti-p-eIF2α S51 | Proteintech | Cat#28740-1-AP |
| Rabbit anti-eIF2α | Proteintech | Cat#11170-1-AP |
| Rabbit anti-p-p38 (T180/Y182) | Cell Signaling | Cat#4511 |
| Rabbit anti-p-MEK (S217/221) | Cell Signaling | Cat#9154 |
| Rabbit anti-p-ERK (T202/Y204) | Cell Signaling | Cat#4370 |
| Rabbit anti-GRP78 | Proteintech | Cat#11587-1-AP |
| Rabbit anti-CHOP | Proteintech | Cat#66741-1-Ig |

| Reagent/resource | Reference or source | Identifier or catalog number |
|---|---|---|
| Rabbit anti-Thiophosphate ester | Abcam | Cat#ab92570 |
| Goat anti-Rabbit IgG (H+L) Secondary Antibody, DyLight™ 800 | Thermo Fisher Scientific | Cat#SA5-35571 |
| Goat anti-Mouse IgG (H+L) Secondary Antibody, DyLight™ 800 | Thermo Fisher Scientific | Cat#SA5-35521 |
| Goat anti-Mouse IgG (H+L) Secondary Antibody, DyLight™ 680 | Thermo Fisher Scientific | Cat#35518 |
| Goat Anti-Rabbit IgG H&L (Alexa Fluor® 488) | Abcam | Cat#ab150077 |
| Goat Anti-Mouse IgG H&L (Alexa Fluor® 488) | Abcam | Cat#ab150113 |
| Goat Anti-Rabbit IgG H&L (Alexa Fluor® 594) | Abcam | Cat#ab150080 |
| Goat Anti-Mouse IgG H&L (Alexa Fluor® 594) | Abcam | Cat#ab150116 |
| DyLight TM 680-coupled Streptavidin Protein | Thermo Fisher Scientific | Cat#21848 |
| **Oligonucleotides and other sequence-based reagents** | | |
| Primers used for ORF amplification | This study | Appendix Table S1 |
| Oligos used for targeting *CAMK2* | This study | Appendix Table S2 |
| siRNA | This study | Appendix Table S2 |
| Primers used for qRT-PCR | This study | Appendix Table S3 |
| **Chemicals, enzymes and other reagents** | | |
| Penicillin–streptomycin | Thermo Fisher Scientific | Cat#15140211 |
| Non-essential amino acids | Thermo Fisher Scientific | Cat#11140076 |
| L-glutamine | Thermo Fisher Scientific | Cat#25030081 |
| Megatran | OriGene | Cat#TT200003 |
| Lipofectamine 2000 | Thermo Fisher Scientific | Cat#11668019 |
| MG132 | Merck Millipore | Cat#474790 |
| KN-93 | Selleck | Cat#S6787 |
| Bortezomib | Topscience | Cat#T2399 |
| Bafilomycin A1 | MCE | Cat#HY-100558 |
| BTdCPU | Selleck | Cat#S0006 |
| ISRIB | Topscience | Cat#T2027 |
| CCT020312 | MCE | Cat#HY-119240 |
| DAPI | Beyotime | Cat#C1006 |
| Hochest 33342 | Solarbio | Cat#C0031 |
| ATP-γ-S | Abcam | Cat#ab138911 |
| p-Nitrobenzyl mesylate | Selleck | Cat#E1248 |
| Azetidine-2-carboxylic acid | MCE | Cat#HY-75308 |
| Wortmannin | MCE | Cat#HY-10197 |
| Protease inhibitor mixture | Thermo Fisher Scientific | Cat#87785 |
| NP-40 Alternative | Merck Millipore | Cat#492016 |

| Reagent/resource | Reference or source | Identifier or catalog number |
|---|---|---|
| Phosphatase inhibitor mixtures | Thermo Fisher Scientific | Cat#78427 |
| Protein-G agarose beads | Merck Millipore | Cat#16-266 |
| Triton X-100 | Merck Millipore | Cat#9410- 1L |
| Phos-tagTM BTL | ApexBio | Cat#BTL104 |
| **Software** | | |
| ImageJ | ImageJ Software | https://imagej.net/software/fiji/ |
| Graphpad Prism 7.0 | GraphPad Software | https://www.graphpad.com/ |
| **Bacterial and virus strains** | | |
| DH5α Chemically Competent Cell | Tsingke | Cat#TSC-C14 |
| BL21 Star(DE3) Chemically Competent Cell | Tsingke | Cat#TSC-E06 |
| **Other** | | |
| Uniclone One Step Seamless Cloning Kit | GeneSand | Cat#SC612 |
| Cell Total RNA Isolation Kit | ForeGene | Cat#RE-03111 |
| PrimeScript 1st strand cDNA Synthesis Kit | TaKaRa | Cat#6110A |
| SYBR Green | Selleck | Cat#B21202 |
| Cell Counting-8 Kit | Dojindo | Cat#CK04 |
| CF488A-ANXA5 apoptosis Kit | Biotium | Cat#30061 |
| Anti-FLAG Magnetic Beads | MCE | Cat#HY-K0207 |
| Click-It HPG Alexa Fluor 488 Protein Synthesis Assay kit | Thermo Fisher Scientific | Cat#C10428 |
| MagStrep "type3" XT Beads | IBA | Cat#2-4090-002 |
| Anti-Myc Magnetic Beads | MCE | Cat#HY-K0206 |

## Cell culture and transfection

HEK293, HeLa and MDA-MB-231 cells were purchased from ATCC. U87MG, U251, LN229 cells and GL261 cells were purchased from OriCell (Guangzhou, China). All the cell lines were authenticated and tested for contamination. LN229 cells were cultured in DMEM medium containing 5% fetal bovine serum (FBS, HyClone, SH30071.03) and 1% penicillin–streptomycin (Thermo Fisher Scientific, 15140211), and other cells were cultured in DMEM medium containing 10% FBS, 1% penicillin–streptomycin, 1× non-essential amino acids (NEAAs, Thermo Fisher Scientific, 11140076) and 2 mM L-glutamine (Thermo Fisher Scientific, 25030081). Cells were cultured in a humidified incubator at 37 °C and 5% $CO_2$. Plasmid transfection was carried out with Megatran (OriGene, TT200003) for HEK293 and HeLa cells, while Lipofectamine 2000 (Thermo Fisher Scientific, 11668019) was utilized for other cells. All siRNA transfections were carried out with Lipofectamine 2000. The transfections were performed according to the manufacturer's instructions.

## Construction of plasmids

Flag- or Myc-tagged BAG3 has been described in previous studies (Zhang et al, 2022b). To construct Flag- or Myc-tagged CaMKIIβ/δ, HSPB8, and HRI expression plasmids, the ORF fragments were amplified from the genes by PCR and cloned into pcDNA3.1 with indicated tags through Uniclone One Step Seamless Cloning Kit (GeneSand, Beijing). Meanwhile, ER-EGFP was constructed by fusing the ER localization signal peptide (MEGKWLLCMLLVLGTAIVE) and ER resident signal sequence (KDEL) at the N- and C-terminus of EGFP, respectively. The corresponding DNA coding sequence was cloned into the pcDNA3.1 vector. To generate mutated BAG3 and CaMKIIβ/δ, the DNA fragments carrying the mutation were obtained by PCR and then cloned into the pcDNA3.1 vector. Appendix Table S1 lists the primer sequence information used for ORF amplification. To generate BAG3 prokaryotic expression plasmids, the wild-type or mutated BAG3 were cloned into pET28a expression vector, with Twin strep tag (IBA) at C-terminal. shRNAs against *CAMK2A/2B/2 G/2D* were expressed in the pLKO.1 vector by cloning annealed oligonucleotides between EcoRI and AgeI sites. Appendix Table S2 lists the targeting sequences. DNA sequencing was utilized to validate all the plasmids.

## Chemical reagents

Herein, the following chemical reagents were used: MG132 (Merck Millipore, 474790), KN-93 (Selleck, S6787), Bortezomib (Topscience, T2399), Bafilomycin A1 (MCE, HY-100558), BTdCPU (Selleck, S0006), ISRIB (Topscience, T2027), CCT020312 (MCE, HY-119240), DAPI (Beyotime, C1006), Hochest 33342 (Solarbio, C0031), ATP-γ-S (Abcam, ab138911), and p-Nitrobenzyl mesylate (PNBM, Selleck, E1248), Azetidine-2-carboxylic acid (MCE, HY-75308), Wortmannin (MCE, HY-10197).

## Gene knockdown

To knock down *CAMK2* expression, the plasmid mixture in equal of shCAMK2s was transfected into HEK293 or HeLa cells. Whereas to knock down PERK, PKR, GCN2, and HRI, the siRNAs corresponding indicated genes were transfected into cells. The following experiments were performed after transfection for 48 h. Appendix Table S2 lists the siRNA sequences.

## Generation of knockout cell lines

The BAG3 knockout cell lines were constructed from HEK293 and HeLa cells using a CRISPR/Cas9 system, as described previously (Zhang et al, 2018). Briefly, two gRNA expression plasmids were co-transfected with the Cas9 expression vector into cells. After 24 h, the cells were diluted and reseeded into 96-well plates to isolate the single colonies. The BAG3 knockout cell lines were identified by western blotting assay. The guide RNA sequences targeting to BAG3 gene were: 5′-tgcaggtggcgtccggcaa-3′ and 5′-agactc-catcctctgccaa-3′.

## Protein extraction

The total protein extract was obtained by homogenizing the cells in a 1× SDS sample buffer. The NP-40-soluble and -insoluble protein fractions from the cells were prepared as previously described

(Zhang et al, 2022a; Zhang et al, 2018). Briefly, the cells were lysed with NP-40 alternative cell lysis buffer (50 mM Tris-HCl, 150 mM NaCl, pH 8.0, 1 mM EDTA, 1% NP-40 Alternative (Merck Millipore, 492016)) containing protease inhibitor mixture (Thermo Fisher Scientific, 87785) on ice for 30 min. Subsequently, the lysates were centrifuged at 14,000×$g$ for 15 min at 4 °C. The supernatant was collected as NP-40 soluble component, and the pellet was solubilized in 1× SDS sample buffer after being washed with PBS three times. Then, the pellet was labeled as insoluble fractions. All the protein samples were boiled and then resolved in an SDS-PAGE.

### Immunoprecipitation

To conduct the immunoprecipitation experiment, the cells were lysed with NP-40 alternative cell lysis buffer containing protease (Thermo Fisher Scientific, 87785) and phosphatase inhibitor mixtures (Thermo Fisher Scientific, 78427) on ice for 30 min. After centrifuging at 14,000× $g$ for 15 min at 4 °C, the supernatant of lysates was collected. Subsequently, primary antibodies, including anti-CaMKII (cell Signaling, 4436), anti-FLAG (Prospec, ANT-146-b), anti-Myc (Biolegend, MMS-150R), anti-BAG3 (Proteintech, 10599-1-AP), anti-HRI (Proteintech, 20499-1-AP), and anti-GFP (Proteintech, 66002-1-Ig), were added and incubated for 2 h or overnight while rotating at 4 °C. The lysates were further incubated with Protein-G agarose beads (Merck Millipore, 16-266) for 1 h at 4 °C, and then washed three times with NP-40 alternative cell lysis buffer. The beads were resuspended in 1× SDS sample buffer and boiled for 5 min.

### Western blot analysis

After boiling, the protein samples were separated by SDS-PAGE and transfected to the Immunobilon-FL PVDF membrane (Merck Millipore, IPFL00010). After blocking with 5% BSA in TBST, the membrane was incubated with primary antibodies overnight at 4 °C. Then, the membrane was washed thrice with TBST and incubated with secondary antibodies for 2 h in darkness. After being washed three times with TBST, the membrane was analyzed with the Li-Cor Odyssey Clx Infrared Imaging System (LI-COR Biotechnology, Lincoln, NE, USA). Quantitative analysis was performed using ImageJ.

### Immunostaining

The immunostaining was performed as described earlier (Zhang et al, 2022a). Briefly, after treatment, the cells that were grown onto glass coverslips and placed in 24- or 48-well plates were fixed using 4% paraformaldehyde for 15 min and then permeabilized with 0.2% Triton X-100 (Merck Millipore, 9410- 1 L) for 15 min. After being blocked with 5% goat serum (Jackson ImmunoResearch Laboratories, 005-000-12) for 1 h, the cells were incubated with primary antibodies for 2 h at room temperature or overnight at 4 °C. The cells were washed and incubated with appropriate secondary antibodies for 1 h at room temperature. After being stained with 2-(4-Amidinophenyl)-6-indolecarbamidine dihydrochloride (DAPI) solution, the coverslips were mounted on slides for fluorescence microscopy. Images were taken by a fluorescent microscope (Nikon Eclipse 80i equipped with Nikon PLAN FLUOR ×40 objective) or

Nikon confocal microscope (Nikon, N-STORM & A1). Photographic images and fluorescence intensities were resized, measured, and analyzed using ImageJ software.

### Real-time RT-qPCR

Total cellular RNA was extracted with Cell Total RNA Isolation Kit (ForeGene, RE-03111) following the manufacturer's instructions. Subsequently, complementary DNAs (cDNAs) were synthesized using PrimeScript 1st strand cDNA Synthesis Kit (TaKaRa, 6110A) according to the manufacturer's instructions. cDNA was analyzed by qPCR with SYBR Green (Selleck, B21202) and gene-specific primers. Appendix Table S3 lists the PCR primer sequences.

### Transmission electronic microscopy

Briefly, the cells were treated with indicated drugs and then prefixed with a 3% glutaraldehyde, postfixed in 1% osmium tetroxide, dehydrated in series acetone, infiltrated in Epox 812 for a longer, and eventually embedded. The semithin sections were stained with methylene blue, and the ultrathin sections were cut with a diamond knife and stained with uranyl acetate and lead citrate. Sections were examined with JEM-1400-FLASH Transmission Electron Microscope.

### Cell viability assay and Annexin-V staining

For cell viability assay, the cells were seeded into 96-well plates at a $3 \times 10^3$/well density to conduct the cell viability assay. After 24 h, the cells were subsequently treated with the indicated reagent. Following the protocols, cell viability was measured using Cell Counting-8 Kit (Dojindo, Kumamoto, Japan). The efficacy of drugs on cell growth was normalized to untreated control. Best-fit curve was generated in GraphPad Prism (inhibitor versus response-variable slope (four parameters)). For Annexin-V staining, cells treated with the specified reagents were harvested using 0.05% trypsin, washed twice with PBS, and stained with CF488A-ANXA5 following the manufacturer's instructions. After staining, the cells were washed twice with binding buffer and mounted onto slides. Images were captured using a Nikon Eclipse 80i fluorescence microscope with a Nikon PLAN FLUOR 40× objective.

### Identification of phosphorylation sites by mass spectrometry

HEK293 cells were cultured in two 10-cm dishes and transfected with BAG3-FLAG as well as Myc vector, CaMKIIβ(D-S)-Myc, and CaMKIIδ(D-S)-MYC, respectively. After the cells were transfected for 24 h, they were treated with the indicated regent. The cells were lysed with NP-40 alternative cell lysis buffer containing protease (Thermo Fisher Scientific, 87785) and phosphatase inhibitor mixtures (Thermo Fisher Scientific, 78427) on ice for 30 min and centrifuged at 14,000× $g$ for 15 min at 4 °C. For the supernatant, BAG3-FLAG was immunopurified with anti-FLAG Magnetic Beads (MCE, HY-K0207). Immunoprecipitated proteins were separated by SDS-PAGE and identified by staining with 0.25% Coomassie brilliant blue R250. The BAG3-FLAG protein bands were cut out and digested with trypsin at 37 °C overnight. The tryptic peptides were dissolved in 0.1% formic acid (solvent A) and directly loaded

onto a homemade reversed-phase analytical column (15 cm length, 75 µm i.d.). The gradient increased from 6 to 23% solvent B (0.1% formic acid in 98% acetonitrile) over 16 min, 23-35% in 8 min and climbing to 80% in 3 min, and then holding at 80% for the last 3 min. The flow rate remained constant at 400 nl/min on an EASY-nLC 1000 UPLC system. The peptides were subjected to an NSI source followed by tandem mass spectrometry (MS/MS) in Q Exactive™ Plus (Thermo) coupled online to the UPLC. The electrospray voltage applied was 2.0 kV. The $m/z$ scan range was 350 to 1800 for a full scan, and the intact peptides were detected in the Orbitrap at a resolution of 70,000. Peptides were then selected for MS/MS using NCE setting at 28, and the fragments were detected in the Orbitrap at a resolution of 17,500. A data-dependent procedure alternated between one MS scan and 20 MS/MS scans with 15.0 s dynamic exclusion. Automatic gain control (AGC) was set at 5E4. The resulting MS/MS data were processed using Proteome Discoverer 2.4. Tandem mass spectra were searched against the Uniport protein database. Trypsin was specified as a cleavage enzyme allowing up to 2 missing cleavages. The mass error was set to 10 ppm and 0.02 Da for precursor and for fragment ions, respectively. Phospho-(S/T/Y) were chosen as variable modifications. Peptide confidence was set at high, and peptide ion score was set at >20.

## Phos-tag based immunoblotting

The Phos-Tag solution (469 µl TBST, 10 µl Phos-tagTM BTL (ApexBio, BTL104)), 20 µl $ZnCl_2$ (10 mM), and 1 µg DyLight TM 680-coupled Streptavidin Protein (Thermo Fisher Scientific, 21848) were incubated in darkness at room temperature for 30 min and then concentrated with 30 K centrifugal filter device. The phos-Tag complex was resuspended in a 30 mL TBST solution. The cells were lysed with NP40 alternative cell lysis buffer containing protease and phosphatase inhibitor mixtures on ice for 30 min and centrifuged at $14,000 \times g$ for 15 min at 4 °C. The target proteins were immuno-precipitated with indicated primary antibodies and subjected to SDS-PAGE. The proteins were then transferred to the Immunobilon-FL PVDF membrane. After the membranes were washed three times with TBST, they were incubated with Phos-Tag complex solution for an hour in the dark. Li-Cor Odyssey Clx Infrared Imaging System was utilized to analyze the results.

## Protein translation assay

Protein synthesis was analyzed using the Click-It HPG Alexa Fluor 488 Protein Synthesis Assay kit (Thermo Fisher Scientific, C10428). The cells were grown onto glass coverslips placed on 48-well plates. After treatment, the protein synthesis assay was performed following the manufacturer's protocol. Images were captured with a fluorescent microscope (Nikon Eclipse 80i) equipped with Nikon PLAN FLUOR ×20 objective. The immuno-fluorescence intensities were measured and analyzed using ImageJ software. For each experiment, three fields containing more than 50 cells each were selected for measurement. The fluorescence value for each group was determined as the ratio of total fluorescence intensity to the number of cells. Within each group, the relative fluorescence intensity values were normalized to untreated cells.

## Protein purification from *E. coli*

pET28a-BAG3-Twin Strep (WT or mutants) were transformed into *E. coli* BL21 (DE3) cells. Subsequently, the colonies were cultured in an LB medium containing 50 µg/mL kanamycin. Until $OD_{600}$ reached ~0.6, the expression was induced by 0.5 mM IPTG for 4 h at 37 °C. Bacterial cells were harvested and resuspended in TBS (50 mM Tris, 150 mM NaCl, pH 8.0) containing protease inhibitor PMSF and then lysed using sonication. After being centrifuged at $14,000 \times g$ for 15 min, the soluble proteins were purified with MagStrep "type3" XT Beads (IBA, 2-4090-002). Purified proteins were eluted using Biotin elution buffer (50 mM Tris-HCl, 75 mM NaCl, 1 mM EDTA, and 10 mM D-biotin) and then concentrated with a 30 K centrifugal filter device. The results were validated using SDS-PAGE.

## In vitro kinase assay

Briefly, two 10-cm dishes of HEK293 cells were transfected Myc vector or the activation mutants of CaMKIIα/β. After 24 h of expression, the cells were lysed with NP-40 alternative cell lysis buffer containing protease inhibitor mixture on ice for 30 min and centrifuged at $14,000 \times g$ for 15 min at 4 °C. The supernatant fractions were incubated with anti-Myc Magnetic Beads (MCE, HY-K0206) for 2 h at 4 °C. Then, the beads were washed three times with NP-40 alternative cell lysis buffer, once with the kinase assay buffer (40 mM HEPES, 50 mM KAc, and 5 mM MgCl2, pH 7.5). Purified kinases and BAG3 were mixed with ATP-γ-S (500 µM) in kinase assay buffer and incubated for 30 min at 30 °C. The reaction was stopped using 20 mM EDTA. Subsequently, the samples were incubated with 2.5 mM PNBM for 1 h at 25 °C. Western blot assay was conducted to analyze the phosphorylation using antibodies against thiophosphate ester.

## Tumor xenograft analysis

All animal experiments followed ethical regulations and were approved by the Subcommittee on research animal care of Western China Hospital of Sichuan University (Approval No. 20220111006). Nude mice (nu/nu, 6-week-old females) were injected subcutaneously with $5 \times 10^6$ MDA-MB-231 cells. The tumors were allowed to reach ~ 50–100 mm³, at which time drug treatment was administered accordingly. The mice were randomly divided into four groups: control (saline), Bortezomib (0.5 mg/kg, intraperitoneally, once every three days), KN-93 (10 mg/kg, intraperitoneally, once every three days), and a combination of Bortezomib and KN-93. The tumor was measured using digital calipers using the formula: tumor volume = 1/2 (length × width²).

## Statistical analysis

Statistical analyses were performed using Graph Prim 7.0. Data were reported as mean ± SEM of three or more biological replicates. Representative images for Immunoblotting, immunostaining, TEM were shown and independently repeated at least three times with a similar tendency unless indicated otherwise described in figure legends. No statistical methods were used to estimate the sample size. Comparisons between individual data points were made as

described in figure legends. Differences were considered statistically significant when $P < 0.05$.

## Data availability

The mass spectrometry proteomics data have been deposited to the ProteomeXchange Consortium via the PRIDE (Perez-Riverol et al, 2022) partner repository with the dataset identifier PXD054722.

The source data of this paper are collected in the following database record: biostudies:S-SCDT-10_1038-S44319-024-00248-w.

## Peer review information

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

## Acknowledgements

This research was supported by the National Natural Science Foundation of China (grant numbers: 32000533, 82073059); 1.1.3.5 Project for Disciplines of excellence, West China Hospital, Sichuan University (grant number: ZYGD18005).

## Author contributions

**Chenliang Zhang**: Conceptualization; Data curation; Funding acquisition; Investigation; Writing—original draft; Project administration. **Huanji Xu**: Resources; Investigation; Methodology; Writing—review and editing. **Qiulin Tang**: Investigation; Methodology. **Yichun Duan**: Investigation. **Hongwei Xia**: Investigation. **Huixi Huang**: Methodology. **Di Ye**: Methodology. **Feng Bi**: Conceptualization; Supervision; Funding acquisition; Project administration; Writing—review and editing.

Source data underlying figure panels in this paper may have individual authorship assigned. Where available, figure panel/source data authorship is listed in the following database record: biostudies:S-SCDT-10_1038-S44319-024-00248-w.

## Disclosure and competing interests statement

The authors declare no competing interests.

# Expanded View Figures

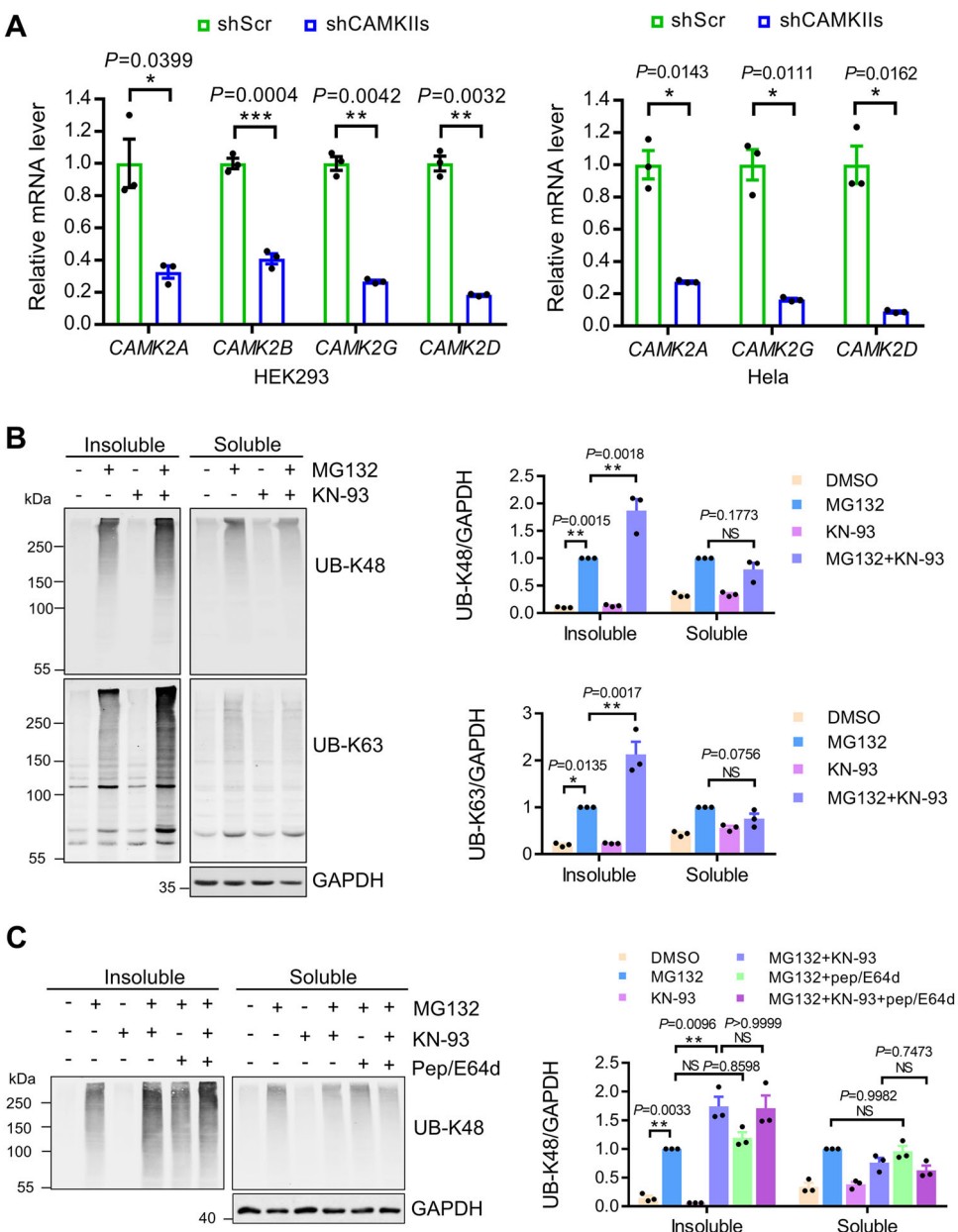

Figure EV1.  CaMKII regulates the synthesis of aggregated ubiquitinated proteins in response to proteasome inhibition, related to Fig. 1.

(A) Confirmation of *CAMK2* knockdown in HEK293 and HeLa cells. After transfected with indicated plasmids for 48 h, the mRNA levels of *CAMK2* were analyzed by qRT-PCR. Data are mean ± SEM of biological replicates ($n = 3$). The P value was determined by Student's *t*-test (two-sided). *$P < 0.05$; **$P < 0.01$; ***$P < 0.001$.
(B) Immunoblotting and quantification of ubiquitin (UB-K48 and UB-K63) in NP-40-soluble and -insoluble fractions of HEK293 cells treated with indicated drugs (MG132 (1 μM), KN-93 (10 μM)) for 14 h. Data are mean ± SEM of biological replicates ($n = 3$). (C) Immunoblotting and quantification of UB-K48 in NP-40-soluble and -insoluble fractions of HEK293 cells treated with indicated drugs (MG132 (1 μM), KN-93 (10 μM), Pepstatin A (Pep, 25 μg/ml)/E64d (25 μg/ml)) for 14 h. Data are mean ± SEM of biological replicates ($n = 3$). For (B, C), the P value was determined by a one-way ANOVA analysis. NS not significant; *$P < 0.05$; **$P < 0.01$.

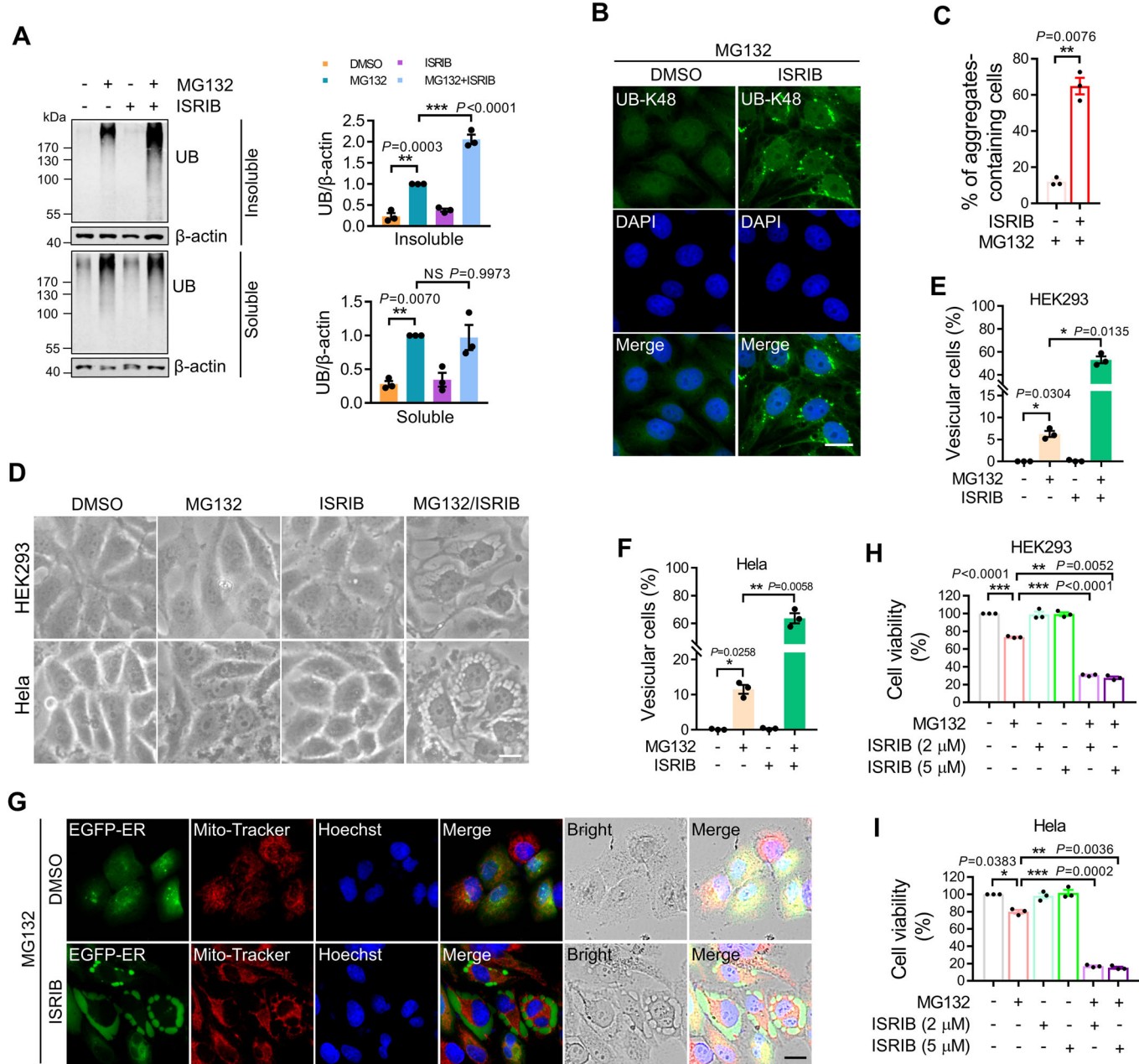

**Figure EV2. Inhibiting eIF2α phosphorylation aggravates the proteasome inhibitor-induced proteotoxicity, related to Fig. 3.**

(A) Immunoblotting and quantification of indicated proteins in NP-40-soluble and -insoluble fractions of HEK293 cells treated with indicated drugs (MG132 (1 μM), ISRIB (5 μM)) for 14 h. Data are mean ± SEM of biological replicates ($n = 3$). (B) Representative UB-K48 staining images of HeLa cells treated with indicated drugs (MG132 (0.5 μM), ISRIB (5 μM)) for 14 h. Scale bar: 20 μm. (C) Quantitative analysis of results in (B). Data are mean ± SEM of biological replicates ($n = 3$). At least 50 cells were randomly selected from each group to score. (D) Cell morphology imaging of HEK293 and HeLa cells treated with indicated drugs (MG132 (1 μM), ISRIB (5 μM)) for 24 h. Scale bar: 10 μm. (E, F) Quantitative analysis of results in (D). Data are mean ± SEM of biological replicates ($n = 3$). At least 50 cells were randomly selected from each group to score. (G) Representative images of HeLa cells transfected with ER-EGFP and treated with indicated drugs (MG132 (1 μM), ISRIB (5 μM)) for 24 h. Mitochondria and nuclei were stained with Mito-Tracker (red) and Hoechst (blue), respectively. Scale bar: 20 μm. (H, I) Cell viability of HEK293 and HeLa cells treated with indicated drugs (MG132 (1 μM), ISRIB (2 μM, 5 μM)) for 48 h. Data are mean ± SEM of biological replicates ($n = 3$). For (C), the $P$ value was determined by Student's $t$ test (two-sided). **$P < 0.01$. For (A, E, F, H, I), the $P$ value was determined by a one-way ANOVA analysis. NS not significant; *$P < 0.05$; **$P < 0.01$; ***$P < 0.001$.

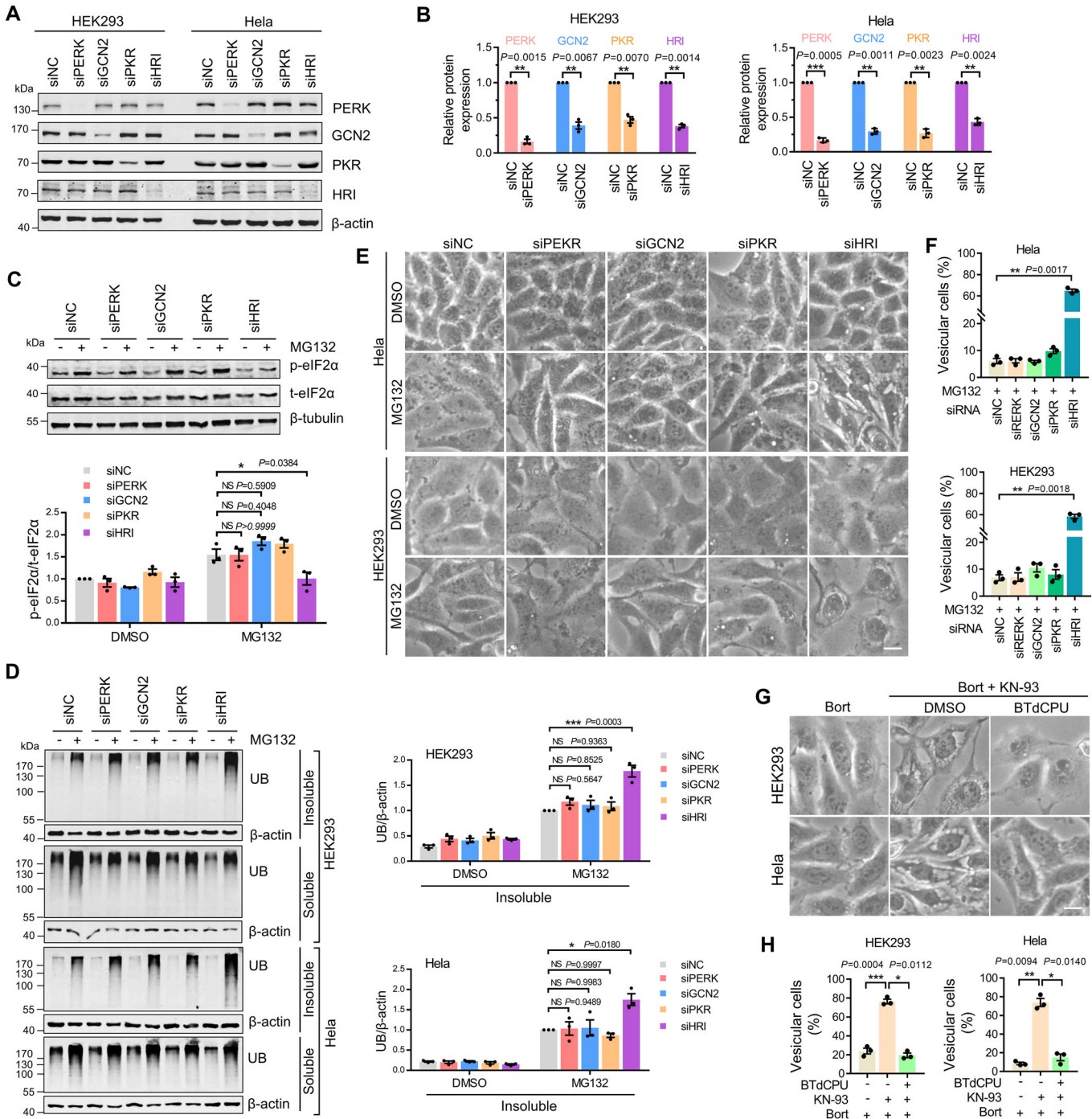

**Figure EV3. HRI is critical for eIF2α activation and proteotoxicity reduction during proteasome dysfunction, related to Fig. 3.**

(A) Confirmation of EIF2AKs knockdown in HEK293 and HeLa cells. After transfected with indicated siRNAs, the whole cell lysates were subjected to western blot analysis with indicated antibodies. (B) Quantitative analysis of results in (A). Data are mean ± SEM of biological replicates ($n = 3$). (C) Immunoblotting and quantification of p-eIF2α (S51) in HeLa cells transfected with indicated siRNAs and treated with MG132. Data are mean ± SEM of biological replicates ($n = 3$). (D) Immunoblotting and quantification of UB in NP-40-soluble and -insoluble fractions of HEK293 and HeLa cells transfected with indicated siRNAs and treated with MG132. Data are mean ± SEM of biological replicates ($n = 3$). (E) Cell morphology imaging of HEK293 and HeLa cells transfected with indicated siRNAs and treated with MG132. Scale bar: 10 μm. (F) Quantitative analysis of results in (E). Data are mean ± SEM of biological replicates ($n = 3$). At least 50 cells were randomly selected from each group to score. (G) Cell morphology imaging of HEK293 and HeLa cells treated with indicated drugs (Bortezomib (1 μM), KN-93 (10 μM), BTdCPU (0.5 μM)) for 24 h. Scale bar: 10 μm. (H) Quantitative analysis of results in (G). Data are mean ± SEM of biological replicates ($n = 3$). At least 50 cells were randomly selected from each group to score. For (B), the P value was determined by Student's t test (two-sided). **$P < 0.01$; ***$P < 0.001$. For (C, D, F, H), the P value was determined by a one-way ANOVA analysis. NS not significant; *$P < 0.05$; **$P < 0.01$; ***$P < 0.001$.

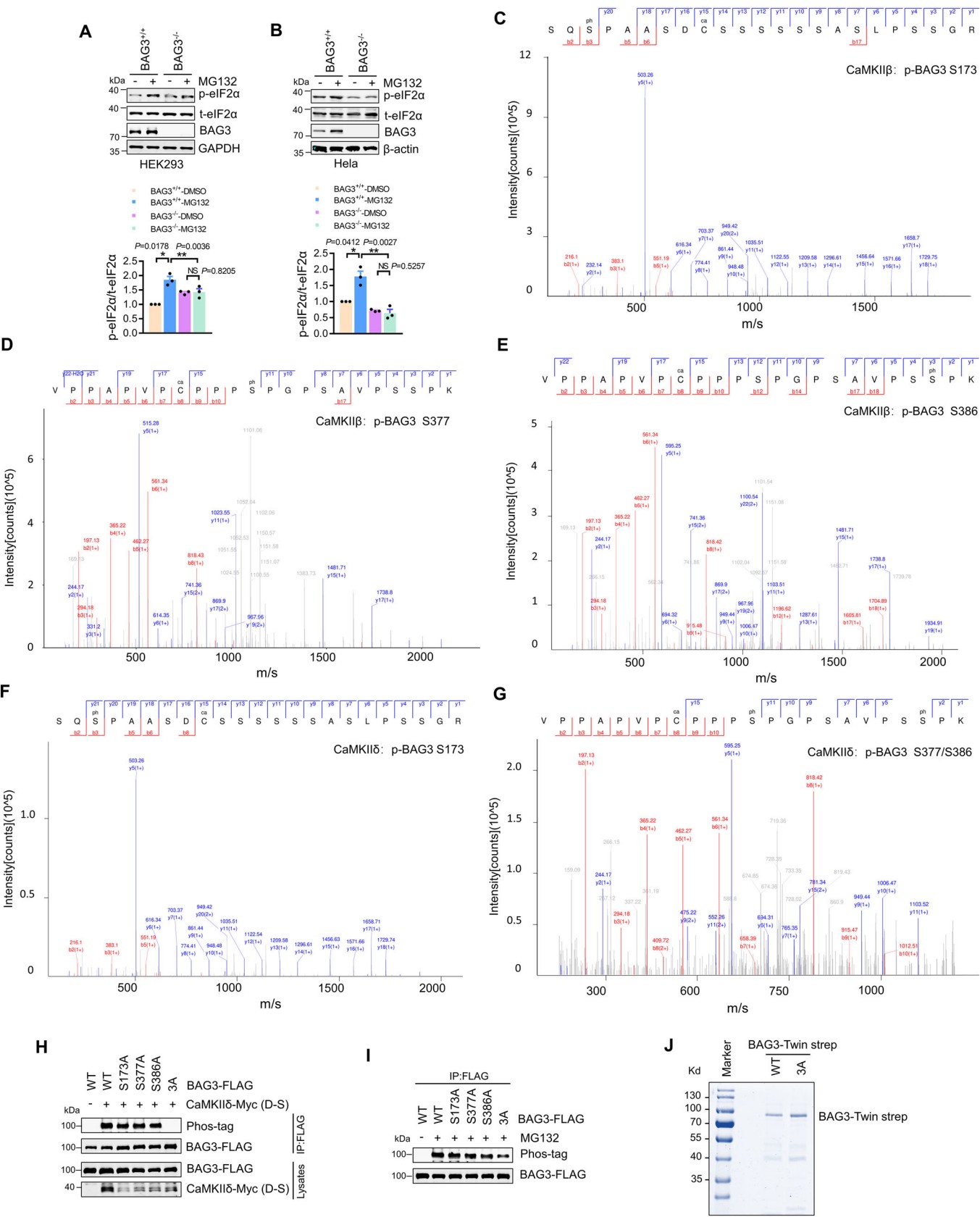

**Figure EV4.  CaMKII phosphorylates BAG3, related to Fig. 4.**

(**A**, **B**) Immunoblotting and quantification of p-eIF2α (S51) in BAG3 knockout cells treated with 1 μM MG132 for 14 h. Data are mean ± SEM of biological replicates ($n = 3$). The $P$ value was determined by Student's $t$ test (two-sided). NS not significant; *$P < 0.05$; **$P < 0.01$. (**C–G**) Mass spectrum of the phosphopeptide containing the S173/S377/S386 phosphorylated residue of BAG3. (**H**, **I**) Phosphorylation level of re-expressed BAG3 (WT or mutants) in BAG3$^{-/-}$ HEK293 cells treated with MG132 (**I**) or not (**H**). (**J**) Confirmation of purified BAG3-Twin Strep (WT or 3A) by coomassie blue staining.

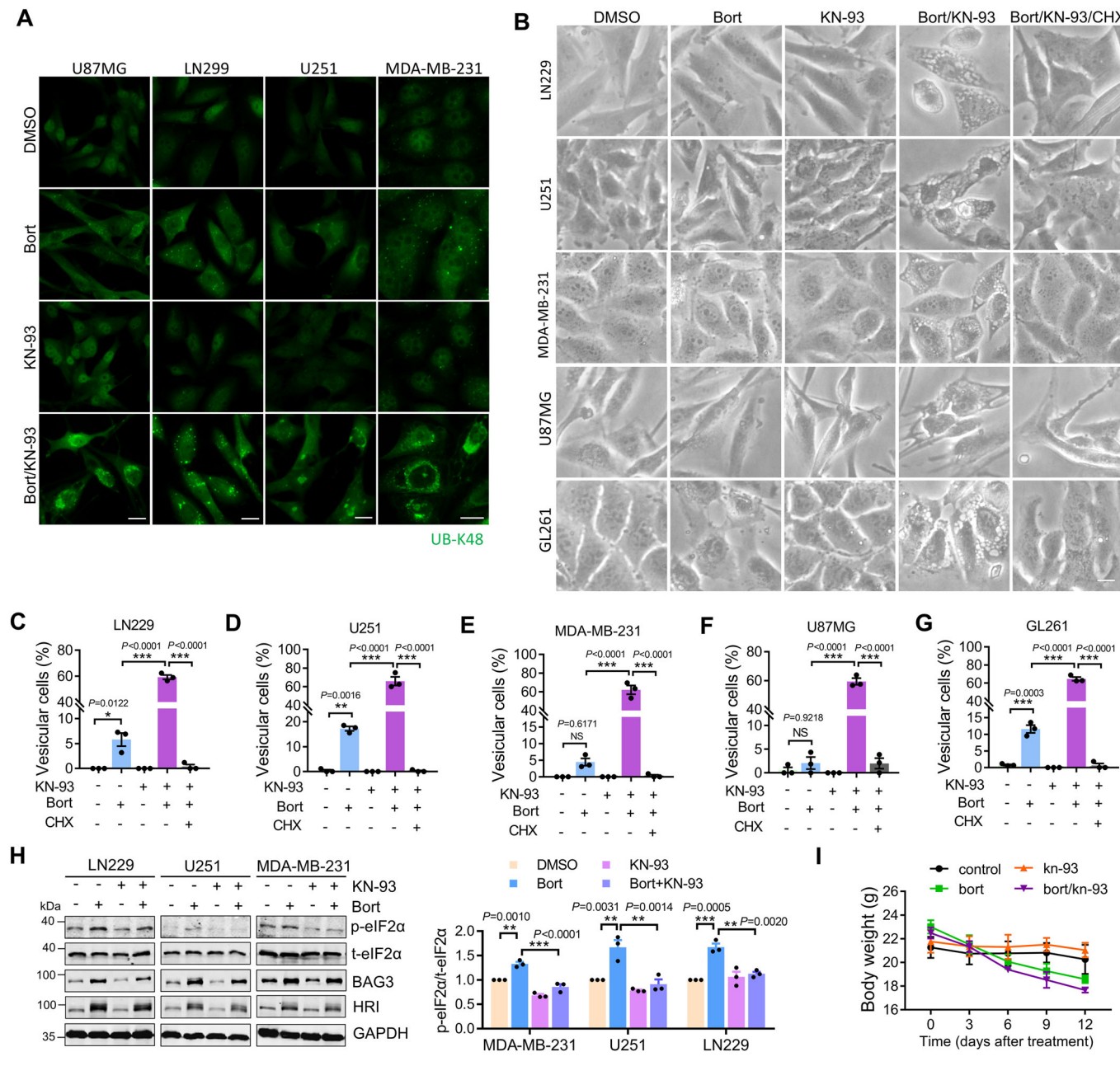

**Figure EV5. Inhibition of CaMKII enhances the anti-tumor activity of proteasome inhibitor, related Fig. 7.**

(A) Representative UB-K48 staining images of tumor cells treated with indicated drugs. Scale bar: 20 μm. (B) Cell morphology imaging of tumor cells treated with indicated drugs (Bortezomib (10 nM), KN-93 (10 μM), CHX (25 μg/ml)) for 24 h. Scale bar: 10 μm. (C–G) Quantitative analysis of results in (B). Data are mean ± SEM of biological replicates (n = 3). At least 50 cells were randomly selected from each group to score. (H) Immunoblotting and quantification of indicated proteins in tumor cells treated with indicated drugs. Data are mean ± SEM of biological replicates (n = 3). (I) The body weight of each group was calculated once time every three days. Data are mean ± SEM of 5 mice. For (C–H), the P value was determined by a one-way ANOVA analysis. NS not significant; *P < 0.05; **P < 0.01; ***P < 0.001.

