## [Peer Review File · EMBO Reports]

CaMKII suppresses proteotoxicity by phosphorylating BAG3 in response to proteasomal dysfunction

Chenliang Zhang, Huanji Xu, Qiulin Tang, YiChun Duan, Hongwei Xia, Huixi Huang, Di Ye, and Feng Bi

Corresponding author(s): Feng Bi (bifeng@scu.edu.cn)

Review Timeline:

Submission Date:	17th Mar 24
Editorial Decision:	5th Apr 24
Revision Received:	2nd Jul 24
Editorial Decision:	26th Jul 24
Revision Received:	9th Aug 24
Accepted:	23rd Aug 24

Transaction Report:

Dear Dr. Bi

Thank you for the submission of your research manuscript to our journal. We have now received the full set of referee reports that is copied below.

As you will see, the referees acknowledge that the findings are interesting and that the conclusions overall supported by the data presented but they also raise a number of concerns and have suggestions how to further strengthen the data that need to be addressed. Two of the referees raised concerns regarding the BAG3 (3D) variant which does not act as constitutive active protein, as one would expect, but rather behaves like the wild-type protein. These data need to be explained and reconciled with the model proposed.

Given these constructive comments, we would like to invite you to revise your manuscript with the understanding that the referee concerns (as detailed above and in their reports) must be fully addressed and their suggestions taken on board. Please address all referee concerns in a complete point-by-point response. Acceptance of the manuscript will depend on a positive outcome of a second round of review. It is EMBO Reports policy to allow a single round of revision only and acceptance or rejection of the manuscript will therefore depend on the completeness of your responses included in the next, final version of the manuscript.

We realize that it is difficult to revise to a specific deadline. In the interest of protecting the conceptual advance provided by the work, we recommend a revision within 3 months (July 5th). Please discuss the revision progress ahead of this time with the editor if you require more time to complete the revisions.

I am also happy to discuss the revision further via e-mail or a video call, if you wish.

*****IMPORTANT NOTE:

We perform an initial quality control of all revised manuscripts before re-review. Your manuscript will FAIL this control and the handling will be delayed IN CASE the following APPLIES:

- 1) A data availability section providing access to data deposited in public databases is missing. If you have not deposited any data, please add a sentence to the data availability section that explains that.
- 2) Your manuscript contains statistics and error bars based on $n=2$. Please use scatter blots in these cases. No statistics should be calculated if $n=2$.

When submitting your revised manuscript, please carefully review the instructions that follow below. Failure to include requested items will delay the evaluation of your revision.*****

- 1) a .docx formatted version of the manuscript text (including legends for main figures, EV figures and tables). Please make sure that the changes are highlighted to be clearly visible.
- 2) individual production quality figure files as .eps, .tif, .jpg (one file per figure). Please download our Figure Preparation Guidelines (figure preparation pdf) from our Author Guidelines pages <https://www.embopress.org/page/journal/14693178/authorguide> for more info on how to prepare your figures.
- 3) a .docx formatted letter INCLUDING the reviewers' reports and your detailed point-by-point responses to their comments. As part of the EMBO Press transparent editorial process, the point-by-point response is part of the Review Process File (RPF), which will be published alongside your paper.
- 4) a complete author checklist, which you can download from our author guidelines (<<https://www.embopress.org/page/journal/14693178/authorguide>>). Please insert information in the checklist that is also reflected in the manuscript. The completed author checklist will also be part of the RPF.

5) Please note that all corresponding authors are required to supply an ORCID ID for their name upon submission of a revised manuscript (<<https://orcid.org/>>). Please find instructions on how to link your ORCID ID to your account in our manuscript tracking system in our Author guidelines (<<https://www.embopress.org/page/journal/14693178/authorguide#authorshipguidelines>>)

6) We replaced Supplementary Information with Expanded View (EV) Figures and Tables that are collapsible/expandable online. A maximum of 5 EV Figures can be typeset. EV Figures should be cited as 'Figure EV1, Figure EV2' etc... in the text and their respective legends should be included in the main text after the legends of regular figures.

7) Before submitting your revision, primary datasets (and computer code, where appropriate) produced in this study need to be deposited in an appropriate public database (see <<https://www.embopress.org/page/journal/14693178/authorguide#dataavailability>>).

Specifically, we would kindly ask you to provide public access to the mass spectrometry datasets.

The accession numbers and database should be listed in a formal "Data Availability " section (placed after Materials & Method) that follows the model below (see also <<https://www.embopress.org/page/journal/14693178/authorguide#dataavailability>>). Please note that the Data Availability Section is restricted to new primary data that are part of this study.

Data availability

Additional information on source data and instruction on how to label the files are available <<https://www.embopress.org/page/journal/14693178/authorguide#sourcedata>>.

10) Figure legends and data quantification:

- the name of the statistical test used to generate error bars and P values,
- the number (n) of independent experiments (please specify technical or biological replicates) underlying each data point,
- the nature of the bars and error bars (s.d., s.e.m.)
- If the data are obtained from n {less than or equal to} 5, show the individual data points in addition to the SD or SEM.
- If the data are obtained from n {less than or equal to} 2, use scatter blots showing the individual data points.

11) Our journal encourages inclusion of *data citations in the reference list* to directly cite datasets that were re-used and obtained from public databases. Data citations in the article text are distinct from normal bibliographical citations and should directly link to the database records from which the data can be accessed. In the main text, data citations are formatted as follows: "Data ref: Smith et al, 2001" or "Data ref: NCBI Sequence Read Archive PRJNA342805, 2017". In the Reference list, data citations must be labeled with "[DATASET]". A data reference must provide the database name, accession number/identifiers and a resolvable link to the landing page from which the data can be accessed at the end of the reference. Further instructions are available at <<https://www.embopress.org/page/journal/14693178/authorguide#referencesformat>>.

12) All Materials and Methods need to be described in the main text. We would encourage you to use 'Structured Methods', our new Methods format. According to this format, the Methods section should include a Reagents and Tools Table (listing key reagents, experimental models, software and relevant equipment and including their sources and relevant identifiers) followed by a Methods and Protocols section in which we encourage the authors to describe their methods using a step-by-step protocol format with bullet points, to facilitate the adoption of the methodologies across labs. More information on how to adhere to this format as well as downloadable templates (.doc or .xls) for the Reagents and Tools Table can be found in our author guidelines: <<https://www.embopress.org/page/journal/14693178/authorguide#manuscriptpreparation>>.

An example of a Method paper with Structured Methods can be found here:
<<https://www.embopress.org/doi/10.15252/msb.20178071>>.

13) As part of the EMBO publication's Transparent Editorial Process, EMBO Reports publishes online a Review Process File to accompany accepted manuscripts. This File will be published in conjunction with your paper and will include the referee reports, your point-by-point response and all pertinent correspondence relating to the manuscript.

Yours sincerely,

Referee #1:

The manuscript titled "CaMKii suppresses proteotoxicity by phosphorylating BAG3 in response to proteasomal dysfunction" focuses on identifying a novel axis of signalling during proteasomal dysfunction and shows the implication of the pathway in regulating viability. The authors primarily report the following findings.

1. Inhibiting proteasome activates CaMKii.
2. This is important for viability and leads to a decrease in ubiquitinated insoluble proteins by decreasing protein synthesis.
3. Protein synthesis is suppressed by phosphorylating eif2a.
4. CaMKii phosphorylates BAG3, which interacts with HRI to activate it.

The work is thorough and connects the dots between two important published observations -

1. HRI senses cytosolic proteostasis to phosphorylate eIF2a.
2. BAG3 activates HRI.

The authors now show what connects proteotoxicity to BAG3 dependent activation of HRI.

While the work is suitable for EMBO reports, there are concerns regarding the interpretations and some of the controls I have pointed out below. I would like these to be addressed (to the extent possible).

Concerns:

Line 113: Is this increase due to proteasome inhibition of accumulation of misfolded proteins? Is there a similar increase upon inhibition of autophagy? Is there a similar increase upon treating cells with conditions that increase cytosolic protein misfolding (AZC for example)?

Line 115: "significantly increased": No numbers report statistical significance in this panel.

Line 116: Fig S1A: Downregulation of CAMKii has been shown by IFC whereas the standard technique is to show either by comparative western blotting or qRT-PCR. Can the authors justify using the less quantitative method to check downregulation by shRNA?

Line 120-121: To state this, the authors also need to show if the puncta are formed when CAMKii is downregulated alone (in the absence of Mg132)

Figure 1E is not called in this section.

Line 122: Fig S1B: Why is beta-actin used as control in some gels while GAPDH is used in others?

Line 133-135: I see a faint increase in Pep/E64d treated cells, specifically in the insoluble fraction. However, there is a strong increase, as expected, when MG132 is present along with Pep/E64D. It will be hard to visually grasp the significance of the upregulation and the validity of the statement without a quantitative graph for the blot.

Line 156-158: I am not an expert in this field so I will not be able to judge the validity of the analysis. To a general scientific audience, the difference in mitochondrial architecture in the Mg132/KN-93 treated cells is not prominent (megamitochondria).

Line 160: CHX cannot be used here as a paraptosis inhibitor as it has been shown by authors to act upstream. It prevents accumulation of Ubq misfolded proteins in the presence of Mg132/KN-93.

Line 171: Fig 2L: Figure 2L has immunoblots for many proteins, the significance of which are not clear. for e.g:

1. ATF6 levels are not indicative of ER stress. The cleavage of ATF6 is. What is being monitored here?
2. GRP78, p-ERK, and p-38 are enhanced similarly in MG132 and KN-93/MG132 in HEL293 cells. Does it mean that the kinase and the ER stress pathways activated upon CAMKii inhibition in the presence of MG132 depends upon other signals?
3. Only p-MEK shows changes consistent to the statement in both cell types.

The figure needs to be elaborated on why these were chosen, or only the relevant blots need to be shown in the main section.

Line 210-211: If a major effector is translation attenuation, why would force activating pERK not help the cell viability? Finally, both should lead to decreased translation and hence should mimic CHX, etc.

Line 233: Fig 4F,G: There is some confusion in the labels in the figure panels. The plasmid added in panel F seems to be CAMKii β , whereas the pulldown is for the delta variant.

Line 291-293: As I understand, the suggested conclusion till this point is: When proteasome is inhibited, BAG3 is phosphorylated by CamKii, this in turn interacts with HRI and activates it. Which in turn decreases translation by phosphorylating eIF2 α .

If this is true then BAG3 (3D) which is a constitutively active BAG3 should interact with HRI in the absence of MG132 (which it doesn't 5D) and lead to eIF2 α phosphorylation independent of MG132 (which it doesn't 5H). When analyzed these along with 5J and 5K, I see that BAG3-3D only works when there is MG132 (independent of CAMKii inhibition 6E). Why does it not work as a dominant mutant?

Line 327: Fig 7H-J: If antioxidants so efficiently reverse the effect, how effective would the combined therapy be in presence of excessive GSSG that may be secreted by cells? (This is likely not the case)

Referee #2:

Zhang and colleagues study the cellular response to proteotoxic stress and identify calcium/calmodulin-dependent protein kinase II (CaMKII) as a regulator of misfolded protein accumulation. Using a combination of depletion of CaMKII expression, CaMKII inhibitors and the CaMKII kinase-dead mutant, the authors first demonstrate the role of CaMKII in the accumulation of ubiquitinated misfolded proteins during cytoplasmic proteotoxicity. Next, they unravel the cascade of its mode of action. They show that CaMKII activity reduces protein synthesis and attenuates proteotoxicity by binding to BAG3, a protein previously described to be involved in the aggregation of ubiquitinated misfolded proteins. The authors identify by mass spectrometry that CaMKII binding to BAG3 leads to BAG3 phosphorylation on Ser173/377/386. This phosphorylation is important for the binding of BAG3 to the stress kinase HRI and its auto-phosphorylation and activation, leading to the phosphorylation of eIF2 α and the suppression of protein synthesis. The author also investigated the possibility of combining the CaMKII inhibitor with proteotoxic drugs to enhance subsequent antitumor activity in a mouse model. In this part of the study, they observed enhanced antitumor activity, suggesting enhanced tumor cell death, paving the way for further in-depth investigations.

The authors present here a comprehensive and well-founded work highlighting an additional regulator, CaMKII, of the previously identified BAG3-HRI-eIF2 α axis and its role in limiting the accumulation of misfolded proteins. Generally, the manuscript is

well-written and the study clearly structured. Results are well controlled and validated by different approaches. Here are a few points to address to strengthen a few remaining aspects.

1. Quantification of all immunoblots should be provided to support the conclusions. Although many differences are marked, differences in especially in ubiquitination, p-eIF2 α levels and phostag signals are sometimes subtle. Also some effects are better visible in 293T cells rather in HeLa cells. It is as important and the quantification provided for immunofluorescence analyses.
2. Measurement of de novo protein synthesis. It is unclear how the percentage of protein translation was estimated and how the cells were analyzed. Is it a measurement of total fluorescence normalized to the control? It seems that the control was untreated cells. The cycloheximide control would be more appropriate and would probably allow greater differences to be observed. How many cells were analyzed? This section of the material and methods could be clearer.
3. Figure S4 identifies HRI as the single stress kinase responsible for activating the integrated stress response in response to proteasome inhibition. Panels h and i show that CCT, an activator of PERK, does not rescue cell viability. It is important to show that this activator is active in this context and to provide immunoblots of activated PERK and p-eIF2 α .

Minor comments:

1. The electron microscopy micrographs are not very important. The main point raised by the authors regarding mitochondria enlargement would require quantification.
2. The molecular weight of HRI varies from approx. 100 kDa (Fig. 4) to 70 kDa. (Fig.S4).
3. Line 200: HRI mediates instead of medicates.
4. Line 209: PERK instead of PEKR.
5. Line 322: cite all panels Fig.7A to 7G.
6. Mass spec results shown in Figure S5 c to f are hardly readable.

Referee #3:

In the present manuscript, a signaling pathway is proposed that is triggered by proteasome inhibition and prevents the accumulation of protein aggregates by attenuating protein synthesis. The authors provide a broad repertoire of biochemical and cell biological experiments to define the constituents of this pathway, which is comprised of CaMKII, BAG3, HRI, and eIF2 α . In general, experiments are convincing and carefully controlled. Some additional controls should be added and original data - and not just quantifications - should be shown in some cases, as outlined below. A major concern regards the 3D variant of BAG3, which is used to demonstrate the CaMKII-dependent phosphorylation and activation of BAG3. In many assays, the 3D variant behaves like wild-type BAG3 and not like a constitutively active protein. This sheds some doubt on the conclusions of the authors regarding the role of BAG3.

Otherwise, it is an important manuscript, as proteasome impairment contributes to a proteostasis imbalance in many diseases and insights into underlying mechanisms are thus of large biomedical relevance. In addition, pharmacologic inhibition of proteasomes is currently explored for tumor therapy and the authors convincingly demonstrate that interference with the identified pathway enhances the anti-tumor activity of proteasome inhibitors.

1. Figure 1A - C: The authors use a phospho-specific antibody to show that phosphorylated CaMKII accumulates under proteasome inhibition. However, it is not clear whether also the total amount of CaMKII increases. Therefore, the authors should also detect total CaMKII with a corresponding antibody.
2. Figure 1D: An anti-CaMKII blot needs to be added to show the level of depletion of the kinase.
3. Figure S1C: Pictures for untransfected controls are missing (original data for grey bars in Figure S1D).
4. Figure 2E: The postulated compound-induced changes in ER and mitochondrial morphology, i.e. ER-derived vacuole expansion and megamitochondria formation, are not evident from the provided pictures. High resolution pictures of all conditions need to be presented and compared.
5. Figure S2D: Original data for detection of Annexin-5 positive cells need to be shown.
6. Figure 2L: The authors conclude from the shown data that MG132 when combined with KN-93 enhanced ER stress. However, ATF6 levels decline upon KN-93 addition. Doesn't this argue against a general enhancement?
7. Figure S4H and I: Labelling should read CCT instead of CTT.
8. Lines 216-219: "We established the BAG3 knockout cell lines and found that deletion of BAG3 decreased the phosphorylation level of eIF2 α and aggravated paraptosis during proteasome inhibition (Fig.S5A and B)." Aggravated paraptosis is not shown in the mentioned figures.

9. Figure S5C-F: Mass spec data are not readable at the resolution of the word file.
10. Figure 4N: The thiophosphate ester panel looks somewhat strange. As AA signals can be seen as prominent faint grey bands - but also lane 4 fades out to the left. Is there a problem with the detection?
11. Figure 5D and E: It is surprising that the 3D variant of BAG3 still shows MG132-induced binding to HRI. If binding of BAG3 to HRI would be dependent on Ser173/377/386 - as claimed by the authors - one would assume that the 3D variant should display constitutive binding to HRI that can no longer be stimulated by MG132.
12. Figure 5G: Again, it is surprising that 3D behaves like wild-type in this assay and not like a constitutive active form that promotes HRI phosphorylation even in the absence of MG132.

Referee #1:

The manuscript titled "CaMKii suppresses proteotoxicity by phosphorylating BAG3 in response to proteasomal dysfunction" focuses on identifying a novel axis of signalling during proteasomal dysfunction and shows the implication of the pathway in regulating viability. The authors primarily report the following findings.

1. Inhibiting proteasome activates CaMKii.
2. This is important for viability and leads to a decrease in ubiquitinated insoluble proteins by decreasing protein synthesis.
3. Protein synthesis is suppressed by phosphorylating eif2a.
4. CaMKii phosphorylates BAG3, which interacts with HRI to activate it.

The work is thorough and connects the dots between two important published observations

-

1. HRI senses cytosolic proteostasis to phosphorylate eIF2a.
2. BAG3 activates HRI.

The authors now show what connects proteotoxicity to BAG3 dependent activation of HRI. While the work is suitable for EMBO reports, there are concerns regarding the interpretations and some of the controls I have pointed out below. I would like these to be addressed (to the extent possible).

Response:

We sincerely appreciate your evaluation of our work and the valuable suggestions you provided, which have significantly contributed to improving the quality of our manuscript and accurately presenting our scientific findings. According to your review comments, we have made every effort to revise our manuscript and would like you to reassess it.

Concerns:

Line 113: Is this increase due to proteasome inhibition of accumulation of misfolded proteins? Is there a similar increase upon inhibition of autophagy? Is there a similar increase upon treating cells with conditions that increase cytosolic protein misfolding (AZC for example)?

Response:

Thank you for raising an excellent point. We have investigated these possibilities. However, as shown in the figure below, we observed that neither AZC nor the autophagy inhibitor Wortmannin could upregulate the levels of phosphorylated CaMKII in cells as MG132 did, suggesting that misfolded proteins and autophagy inhibition are not the triggers for CaMKII activation.

Figure for referees not shown.

* HeLa cells were treated with indicated drugs for 14 h. The whole cell lysates were subjected to western blot analysis with indicated antibodies. AZC, Azetidine-2-carboxylic acid; Wort, Wortmannin.

Line 115: "significantly increased": No numbers report statistical significance in this panel.

Response:

Thank you for your correction. In the revised manuscript, we have supplemented quantitative statistical analysis of the corresponding experimental results (new Fig. 1E).

Line 116: Fig S1A: Downregulation of CAMKii has been shown by IFC whereas the standard technique is to show either by comparative western blotting or qRT-PCR. Can the authors justify using the less quantitative method to check downregulation by shRNA?

Response:

Thank you for your suggestion. In the revised manuscript, we evaluated the knockdown efficiency of shCAMKIIIs on *CAMK2* using qRT-PCR. As shown in Figure EV1, shCAMKIIIs significantly reduced the mRNA levels of all four *CAMK2* isoforms in HEK293 cells and three isoforms in Hela cells, except for *CAMK2B* that was not detected in the qPCR experiment.

Line 120-121: To state this, the authors also need to show if the puncta are formed when CAMKii is downregulated alone (in the absence of Mg132)

Response:

In the revised manuscript, we have added the experimental results of UB-K48 aggregation when CAMKIIIs knockdown alone. As shown in Fig. 1F, we observed that in the absence of MG132, knockdown CAMKIIIs didn't induce the formation of UB-K48 puncta.

Figure 1E is not called in this section.

Response:

We have cited it (new Fig. 1F) in this section.

Line 122: Fig S1B: Why is beta-actin used as control in some gels while GAPDH is used in others?

Response:

Usually, in WB experiments, beta-actin can be used as a loading control for both the soluble and insoluble fractions when mild detergents are used, while GAPDH, due to its high solubility, is typically used only for the soluble fraction. Therefore, in this work, we used beta-actin as the loading control for the insoluble fraction and either beta-actin or GAPDH as the loading control for the soluble fraction. In Fig. EV1B and C, we used GAPDH from the soluble fraction as the loading control for the samples.

Line 133-135: I see a faint increase in Pep/E64d treated cells, specifically in the insoluble fraction. However, there is a strong increase, as expected, when MG132 is present along with Pep/E64D. It will be hard to visually grasp the significance of the upregulation and the validity of the statement without a quantitative graph for the blot.

Response:

We apologize for the confusion caused by our improper expression. Here, we mainly intended to explain that the lysosome inhibitor Pep/E64D does not affect the accumulation of ubiquitinated proteins in the insoluble fraction induced by MG132. This indicates that

aggrephagy is not involved in the accumulation of aggregated ubiquitinated proteins induced by CaMKII inhibition. In the revised manuscript, we have clarified the relevant experimental results and conducted quantitative analysis of the results in Fig. EV1C.

Line 156-158: I am not an expert in this field so I will not be able to judge the validity of the analysis. To a general scientific audience, the difference in mitochondrial architecture in the Mg132/KN-93 treated cells is not prominent (megamitochondria).

Response:

Thank you for timely correcting our improper expression. A similar suggestion was also raised by another reviewer. Our data indeed do not support the conclusion regarding megamitochondria. In the revised manuscript, we removed the statement about "megamitochondria."

Line 160: CHX cannot be used here as a paraptosis inhibitor as it has been shown by authors to act upstream. It prevents accumulation of Ubq misfolded proteins in the presence of Mg132/KN-93.

Response:

This is a great question that helps us interpret the data more reasonably. Paraptosis, as a form of regulated cell death, currently lacks unequivocal markers to distinguish it from other forms of regulated cell death. Researchers mainly identify paraptosis through cell morphology and concurrent cellular stress responses. Excessive accumulation of ubiquitinated proteins is a major inducer of paraptosis, and inhibiting protein synthesis can reduce the accumulation of ubiquitinated proteins, thereby inhibiting paraptosis. We previously described how inhibiting/knocking down CAMKII exacerbates the protein synthesis-dependent accumulation of ubiquitinated proteins. As you pointed out, using CHX to identify paraptosis here is inappropriate. The rescue effect of CHX on the exacerbated cellular damage due to CAMKII inhibition should more objectively indicate that this cellular damage originates from intense protein synthesis. In the revised manuscript, we removed the emphasis on CHX as a paraptosis inhibitor. However, we also examined other features related to paraptosis, such as ROS dependence, enhanced ER stress, and activation of the MAPK signaling pathway, which also indicate that MG132/KN-93 exacerbates cell paraptosis.

Line 171: Fog 2L: Figure 2L has immunoblots for many proteins, the significance of which are not clear. for e.g:

1. ATF6 levels are not indicative of ER stress. The cleavage of ATF6 is. What is being monitored here?
2. GRP78, p-ERK, and p-38 are enhanced similarly in MG132 and KN-93/MG132 in HEL293 cells. Does it mean that the kinase and the ER stress pathways activated upon CAMKII inhibition in the presence of MG132 depends upon other signals?
3. Only p-MEK shows changes consistent to the statement in both cell types.

The figure needs to be elaborated on why these were chosen, or only the relevant blots need to be shown in the main section.

Response:

Thank you for your suggestion. Paraptosis is typically associated with increased ER stress and activation of the MAPK signaling pathway. We examined these pathways mainly to identify paraptosis. We found that compared to MG132 treatment alone, MG132/KN-93 co-treatment reduced the protein levels of full-length ATF6, which does not necessarily indicate increased ATF6 cleavage. In the revised manuscript, we removed the ATF6 detection results, as this does not affect the conclusion that MG132/KN-93 exacerbates ER stress. Other results, such as the upregulation of GRP78 and CHOP protein levels, support this conclusion. As shown in Fig. 2L, MG132/KN-93 significantly upregulated the protein levels of GRP78, CHOP, p-MEK, and p-p38 compared to MG132 alone, indicating that inhibiting CAMKII enhances ER stress and the MAPK signaling pathway, consistent with paraptosis. However, it remains unclear whether these cellular stress responses are causes or manifestations of paraptosis. While the enhancement of the MAPK signaling pathway is a key feature of paraptosis, there is no unified standard for the behavior of specific signaling molecules in this pathway during paraptosis. We examined major members of the MAPK signaling pathway—phosphorylated MEK, ERK, and p38—and found that MG132/KN-93 treatment upregulated phosphorylated MEK and p38 in both HEK293 and HeLa cells, while phosphorylated ERK was only upregulated in HeLa cells. This may reflect different response mechanisms in different cell types. However, this does not affect our conclusion that MG132/KN-93 treatment enhances the MAPK signaling pathway. Additionally, it remains unclear from current data whether the enhancement of ER stress and the MAPK signaling pathway by CAMKII inhibition during proteasome inhibition depends on other signals. We believe that with growing attention to CAMKII in proteasome stress, future research will address this question.

Line 210-211: If a major effector is translation attenuation, why would force activating PERK not help the cell viability? Finally, both should lead to decreased translation and hence should mimic CHX, etc.

Response:

This is a very good question. Another reviewer also suggested that we investigate the effect of activating PERK on eIF2 α activity in MG132/KN-93 treated cells. We found that the PERK-specific agonist CCT020312 can reverse the inhibition of eIF2 α phosphorylation induced by MG132/KN-93, as well as the accumulation of ubiquitinated proteins (Appendix Fig. S1A, B). However, it cannot reverse the cell damage caused by MG132/KN-93 (Appendix Fig. S1C, D). We speculate that activating PERK might trigger other types of cell damage, such as apoptosis. After all, the PERK-eIF2 α -ATF4 signaling axis is a crucial pathway for inducing apoptosis. Interestingly, HRI can also phosphorylate eIF2 α , but this mediates cell survival. This suggests that during proteasome inhibition, PERK- and HRI-mediated eIF2 α phosphorylation have different effects on cell fate, and the underlying molecular mechanisms need to be elucidated in future studies. In the revised manuscript, we have added a discussion on this issue.

Line 233: Fig 4F,G: There is some confusion in the labels in the figure panels. The plasmid added in panel F seems to be CAMKIIbeta, whereas the pulldown is for the delta variant.

Response:

Thank you for your correction. In the revised manuscript we have made corrections.

Line 291–293: As I understand, the suggested conclusion till this point is: When proteasome is inhibited, BAG3 is phosphorylated by CamKii, this in turn interacts with HRI and activates it. Which in turn decreases translation by phosphorylating eIF2a.

If this is true then BAG3 (3D) which is a constitutively active BAG3 should interact with HRI in the absence of MG132 (which it doesn't 5D) and lead to eIF2a phosphorylation independent of MG132 (which it doesn't 5H). When analyzed these along with 5J and 5K, I see that BAG3-3D only works when there is MG132 (independent of CAMKii inhibition 6E). Why does it not work as a dominant mutant?

Response:

This is an excellent question, and another reviewer raised a similar issue. As our results demonstrate, BAG3 (3D) (but not BAG3 (3A)) only works under proteasome inhibition. We speculate that BAG3 phosphorylation might be a necessary condition for the activation of the HRI-eIF2 α signaling pathway, but not a sufficient one. Other signals or molecules activated by proteasome inhibition might also be required in this process. In the revised manuscript, we have thoroughly corrected our description of BAG3's regulatory role in the HRI-eIF2 α signaling pathway to make it more objective and accurate, and we have added a discussion on the related points in the discussion section.

Line 327: Fig 7H–J: If antioxidants so efficiently reverse the effect, how effective would the combined therapy be in presence of excessive GSSG that may be secreted by cells? (This is likely not the case)

Response:

Your concern is well-founded. The high levels of GSH in the tumor microenvironment might indeed be detrimental to the efficacy of drugs that rely on ROS for their cytotoxic effects. However, the elevated levels of ROS present in both tumor cells and the tumor microenvironment also provide favorable conditions for drug efficacy. Thus, the impact of endogenous reductants/oxidants on drug treatment should be considered as a combined effect. Although our *in vivo* experiments (Fig. 7K–M) showed that the combination of Bortezomib and KN-93 significantly inhibited tumor growth, whether the GSH/GSSG in the tumor microenvironment has a counteractive effect on the treatment still needs to be elucidated in future studies.

Referee #2:

Zhang and colleagues study the cellular response to proteotoxic stress and identify calcium/calmodulin-dependent protein kinase II (CaMKII) as a regulator of misfolded protein accumulation. Using a combination of depletion of CaMKII expression, CaMKII inhibitors and the CaMKII kinase-dead mutant, the authors first demonstrate the role of CaMKII in the accumulation of ubiquitinated misfolded proteins during cytoplasmic proteotoxicity. Next, they unravel the cascade of its mode of action. They show that CaMKII activity reduces protein synthesis and attenuates proteotoxicity by binding to BAG3, a protein previously described to be involved in the aggregation of ubiquitinated misfolded

proteins. The authors identify by mass spectrometry that CaMKII binding to BAG3 leads to BAG3 phosphorylation on Ser173/377/386. This phosphorylation is important for the binding of BAG3 to the stress kinase HRI and its auto-phosphorylation and activation, leading to the phosphorylation of eIF2alpha and the suppression of protein synthesis. The author also investigated the possibility of combining the CaMKII inhibitor with proteotoxic drugs to enhance subsequent antitumor activity in a mouse model. In this part of the study, they observed enhanced antitumor activity, suggesting enhanced tumor cell death, paving the way for further in-depth investigations.

The authors present here a comprehensive and well-founded work highlighting an additional regulator, CaMKII, of the previously identified BAG3-HRI-eIF2alpha axis and its role in limiting the accumulation of misfolded proteins. Generally, the manuscript is well-written and the study clearly structured. Results are well controlled and validated by different approaches. Here are a few points to address to strengthen a few remaining aspects.

Response:

We sincerely appreciate your evaluation and positive feedback on our work, as well as your valuable suggestions to further enhance the quality of the manuscript. According to your review comments, we have made every effort to revise our manuscript and would like you to reassess it.

1. Quantification of all immunoblots should be provided to support the conclusions. Although many differences are marked, differences in especially in ubiquitination, p-eIF2alpha levels and phostag signals are sometimes subtle. Also some effects are better visible in 293T cells rather in HeLa cells. It is as important and the quantification provided for immunofluorescence analyses.

Response:

Thank you very much for your suggestion. In the revised manuscript, we have supplemented quantitative analysis for these immunoblots and immunofluorescence.

2. Measurement of de novo protein synthesis. It is unclear how the percentage of protein translation was estimated and how the cells were analyzed. Is it a measurement of total fluorescence normalized to the control? It seems that the control was untreated cells. The cycloheximide control would be more appropriate and would probably allow greater differences to be observed. How many cells were analyzed? This section of the material and methods could be clearer.

Response:

We apologize for the confusion caused by our unclear expression. The data here indeed represent a measurement of fluorescence normalized to the control (untreated cells). For each independent experiment, we analyzed the fluorescence signals of at least 50 cells. In the revised manuscript, we supplemented the detection of cells treated with CHX (new Fig. 1K). However, due to the strong inhibition of protein synthesis, we consider it a reference for experimental validity rather than a suitable normalization control for quantitative analysis. Additionally, we have changed the y-axis label in the original figure to "Normalized Click-iT HPG intensity" for clarity (new Fig. 1K and 5I). We have also added relevant experimental

procedure descriptions in the methods section.

3. Figure S4 identifies HRI as the single stress kinase responsible for activating the integrated stress response in response to proteasome inhibition. Panels h and i show that CCT, an activator of PERK, does not rescue cell viability. It is important to show that this activator is active in this context and to provide immunoblots of activated PERK and p-eIF2 α .

Response:

Great suggestion. Another reviewer also raised a similar concern regarding the inability of activating PERK to rescue the cell damage caused by MG132/KN-93. We examined the effect of CCT on rescuing p-eIF2 α and found that CCT can successfully rescue the inhibition of p-eIF2 α caused by CAMKII inhibition (Appendix Fig. S1A). We also found that CCT can rescue the accumulation of ubiquitinated proteins caused by CAMKII inhibition (Appendix Fig. S1B). However, CCT cannot rescue cell viability (Appendix Fig. S1C, D). We speculate that activating PERK might trigger other types of cell damage, such as apoptosis. After all, the PERK-eIF2 α -ATF4 signaling axis is a crucial pathway for inducing apoptosis. The effects of PERK- and HRI-mediated eIF2 α phosphorylation on cell fate might differ, and the molecular mechanisms need to be elucidated in future studies. In the revised manuscript, we have added a discussion on this issue.

Minor comments:

1. The electron microscopy micrographs are not very important. The main point raised by the authors regarding mitochondria enlargement would require quantification.

Response:

Thank you for pointing this out. Our data indeed do not conclusively support the conclusion of "mitochondria enlargement." In the revised manuscript, we have removed references to "mitochondria enlargement."

2. The molecular weight of HRI varies from approx. 100 kDa (Fig. 4) to 70 kDa. (Fig.S4).

Response:

We have made the correct mark in the revised manuscript.

3. Line 200: HRI mediates instead of medicates.

Response:

It has been modified in the new manuscript.

4. Line 209: PERK instead of PEKR.

Response:

It has been modified in the new manuscript.

5. Line 322: cite all panels Fig.7A to 7G.

Response:

We have cited Fig. 7A to 7G in the new manuscript.

6. Mass spec results shown in Figure S5 c to fare hardly readable.

Response:

In the revised manuscript, we have enlarged these Figures (new Fig. EV5C-G) to make them more readable.

Referee #3:

In the present manuscript, a signaling pathway is proposed that is triggered by proteasome inhibition and prevents the accumulation of protein aggregates by attenuating protein synthesis. The authors provide a broad repertoire of biochemical and cell biological experiments to define the constituents of this pathway, which is comprised of CaMKII, BAG3, HRI, and eIF2a. In general, experiments are convincing and carefully controlled. Some additional controls should be added and original data - and not just quantifications - should be shown in some cases, as outlined below. A major concern regards the 3D variant of BAG3, which is used to demonstrate the CaMKII-dependent phosphorylation and activation of BAG3. In many assays, the 3D variant behaves like wild-type BAG3 and not like a constitutively active protein. This sheds some doubt on the conclusions of the authors regarding the role of BAG3.

Otherwise, it is an important manuscript, as proteasome impairment contributes to a proteostasis imbalance in many diseases and insights into underlying mechanisms are thus of large biomedical relevance. In addition, pharmacologic inhibition of proteasomes is currently explored for tumor therapy and the authors convincingly demonstrate that interference with the identified pathway enhances the anti-tumor activity of proteasome inhibitors.

Response:

We sincerely appreciate your evaluation of our work, as well as your important suggestions to further improve the manuscript. According to your review comments, we have made every effort to revise our manuscript and would like you to reassess it.

1. Figure 1A – C: The authors use a phosphor-specific antibody to show that phosphorylated CaMKII accumulates under proteasome inhibition. However, it is not clear whether also the total amount of CaMKII increases. Therefore, the authors should also detect total CaMKII with a corresponding antibody.

Response:

Thank you for your suggestion. We have supplemented the total CaMKII detection results (new Fig. 1A-C) in the revised manuscript.

2. Figure 1D: An anti-CaMKII blot needs to be added to show the level of depletion of the kinase.

Response:

In the revised manuscript, we have used q-PCR to assess the knockdown efficiency of CaMKIIs (new Fig. EV1A).

3. Figure S1C: Pictures for untransfected controls are missing (original data for grey bars in

Figure S1D).

Response:

Thank you for your correction. In this experiment, we evaluated the impact of CAMKII α expression/high expression (green bars) and no expression/low expression (grey bars) on UB-K48 aggregation within the same transfection well. Upon re-evaluation, we realized that this experimental design cannot strictly distinguish the expression of exogenous CAMKII α . Therefore, we have removed these experimental results from the revised manuscript.

4. Figure 2E: The postulated compound-induced changes in ER and mitochondrial morphology, i.e. ER-derived vacuole expansion and megamitochondria formation, are not evident from the provided pictures. High resolution pictures of all conditions need to be presented and compared.

Response:

Thank you for your reminder. The results of Figure 2E are insufficient to support the "megamitochondria" conclusion, and we have removed the related descriptions from the revised manuscript. However, regarding the "ER-derived vacuole expansion," the accumulation of the ER resident protein marker (ER-EGFP) in vacuoles in Figure 2E still supports this conclusion. In the revised manuscript, we have magnified all images under different treatment conditions to facilitate comparison (new Fig. 2E).

5. Figure S2D: Original data for detection of Annexin-5 positive cells need to be shown.

Response:

We have added the necessary supplements in the revised manuscript (new Fig. EV2D).

6. Figure 2L: The authors conclude from the shown data that MG132 when combined with KN-93 enhanced ER stress. However, ATF6 levels decline upon KN-93 addition. Doesn't this argue against a general enhancement?

Response:

Cleaved ATF6, rather than full-length ATF6, is considered an indicator of ER stress. Here, we found that compared to MG132 treatment alone, MG132/KN-93 co-treatment reduced the protein levels of full-length ATF6. However, this does not strictly represent an increase in ATF6 cleavage. In the revised manuscript, we have removed the ATF6 detection results. This does not affect the conclusion that MG132/KN-93 exacerbates cellular ER stress, as other results (such as the upregulation of GRP78 and CHOP protein levels) also support this conclusion.

7. Figure S4H and I: Labelling should read CCT instead of CTT.

Response:

It has been modified in the new manuscript (new Appendix Fig. S1C, D).

8. Lines 216-219: "We established the BAG3 knockout cell lines and found that deletion of BAG3 decreased the phosphorylation level of eIF2 α and aggravated paraptosis during proteasome inhibition (Fig.S5A and B)." Aggravated paraptosis is not shown in the mentioned figures.

Response:

We apologize for our erroneous description. We have removed the text "and aggravated paraptosis" from the revised manuscript.

9. Figure S5C-F: Mass spec data are not readable at the resolution of the word file.

Response:

In the revised manuscript, we have enlarged these figures (new Fig. EV5C-G) to make them more readable.

10. Figure 4N: The thiophosphate ester panel looks somewhat strange. As AA signals can be seen as prominent faint grey bands - but also lane 4 fades out to the left. Is there a problem with the detection?

Response:

Thank you for your reminder. We have re-examined the immunoblots in Figure 4N and still believe that they objectively reflect the experimental results. In the revision submission, we have uploaded the original WB images for your re-evaluation.

11. Figure 5D and E: It is surprising that the 3D variant of BAG3 still shows MG132-induced binding to HRI. If binding of BAG3 to HRI would be dependent on Ser173/377/386 - as claimed by the authors - one would assume that the 3D variant should display constitutive binding to HRI that can no longer be stimulated by MG132.

12. Figure 5G: Again, it is surprising that 3D behaves like wild-type in this assay and not like a constitutive active form that promotes HRI phosphorylation even in the absence of MG132.

Response to 11 and 12:

Response:

This is an excellent question, and another reviewer raised a similar point. As demonstrated by our results, BAG3 (3D), rather than BAG3(3A), only works under proteasome inhibition. We hypothesize that BAG3 phosphorylation may be a necessary, but not sufficient, condition for the activation of the HRI-eIF2 α signaling pathway. Other signals or molecules activated by proteasome inhibition might also be required in this process. In the revised manuscript, we have thoroughly revised the description of BAG3's role in regulating the HRI-eIF2 α signaling pathway to make it more accurate and objective, and we have added relevant points to the discussion section.

Dear Dr. Bi

Thank you for the submission of your revised manuscript to EMBO reports. We have now received the full set of referee reports that is copied below.

As you will see, both referees are very positive about the study and support publication.

Browsing through the manuscript myself, I noticed a few editorial things that we need before we can proceed with the official acceptance of your study.

- Please reduce the number of keywords to 5.

- Regarding the Author Contributions, we now use CRediT to specify the contributions of each author in the journal submission system. Therefore, please remove the Author Contributions from the manuscript file and make sure that the author contributions in our online manuscript tracking system are correct and up-to-date. The information you specified in the system will be automatically retrieved and typeset into the article. You can enter additional information in the free text box provided, if you wish.

- Please update the references to the alphabetical Harvard style. The abbreviation 'et al' should be used if there are more than 10 authors. You can download the respective EndNote file from our Guide to Authors https://endnote.com/style_download/embo-reports/

- Please note that all corresponding authors are required to supply an ORCID ID for their name upon submission of a revised manuscript (<<https://orcid.org/>>). This information is still missing for Dr. Bi. Please find instructions on how to link your ORCID ID to your account in our manuscript tracking system in our Author guidelines (<<https://www.embopress.org/page/journal/14693178/authorguide#authorshipguidelines>>)

- Animal experiments: please provide the reference number for the approval in the methods section.

- Data availability: please deposit the mass spectrometry data in a public repository and provide the URL linking to the dataset in the Data Availability section.

- Since July we require that all manuscript adhere to our 'Structured Methods' format. According to this format, the Methods section includes a Reagents and Tools Table (listing key reagents, experimental models, software and relevant equipment and including their sources and relevant identifiers) followed by a Methods and Protocols section describing the methods using a step-by-step protocol format. The aim is to facilitate adoption of the methodologies across labs.

In your case this in essence means that we would require you to fill and upload a Reagents and Tools table. Since this table will also include information on antibodies and oligonucleotides, some of the Appendix Tables will be redundant and can be removed.

More information on how to adhere to this format as well as a downloadable template (.docx) for the Reagents and Tools Table can be found in our author guidelines: <https://www.embopress.org/page/journal/14693178/authorguide#structuredmethods>.

An example of a Method paper with Structured Methods can be found here: <https://www.embopress.org/doi/10.15252/msb.20178071>.

- The manuscript sections should be in the following order: Title page - Abstract & Keywords - Introduction - Results - Discussion - Methods - Data Availability - Acknowledgments - Disclosure Statement & Competing Interests - References - Figure Legends - (Main Tables with legends) - Expanded View Figure Legends.

- Please add a callout for Appendix Figure S2 in the text.

- Please note that we can only typeset up to 5 EV figures; you currently have 7. Please either move some of the figures to the Appendix or the main manuscript.

- Our production/data editors have asked you to clarify several points in the figure legends (see below). Please incorporate these changes in the manuscript and return the revised file with tracked changes with your final manuscript submission.

A) Statistical test information. Only p-values that are actually shown in the figure panel(s) should (and must) be defined in the legends, all others should be removed from (or added to) the legend. Moreover, we ask for the specification of exact p-values:

- Please define the annotated p values ***/**/* as well as provide the exact p-values for the same in the legend of figure 7g-j; as appropriate.

- Please note that the exact p values are not provided in the legends of figures 1a-b, d-e, g-h, k-m, o; 2b, d, h-l, 3a, c-e, g, i, k-l; 4h, m; 5d-i; 6a, c, e, g, i, k; 7k-m, o-p; EV 1a-c; EV 2b-c, e; EV 3a, c, e-f, h-i; EV 4b-d, f, h; EV 5a-b; EV 6b, d; EV 7c-h.
- Please indicate the statistical test used for data analysis in the legends of figures 7g-j.
- Please note that in figures 7k-m; there is a mismatch between the annotated p values in the figure legend and the annotated p values in the figure file that should be corrected.

B) Data presentation:

- Please note that the scale bar needs to be defined for figures EV 2d; EV 3d; EV 4e.

- Finally, EMBO Reports papers are accompanied online by

A) a short (1-2 sentences) summary of the findings and their significance,

B) 2-3 bullet points highlighting key results and

C) a schematic summary figure that provides a sketch of the major findings (not a data image).

Please provide the summary figure as a separate file in PNG or JPG format at a size of 550x300-600 pixels (width x height).

Please note that the size is rather small and that text needs to be readable at the final size. Please send us this information along with the revised manuscript.

- On a different note, I would like to alert you that EMBO Press offers a new format for a video-synopsis of work published with us, which essentially is a short, author-generated film explaining the core findings in hand drawings, and, as we believe, can be very useful to increase visibility of the work. This has proven to offer a nice opportunity for exposure i.p. for the first author(s) of the study. Please see the following link for representative examples and their integration into the article web page:

<https://www.embopress.org/doi/full/10.15252/emboj.2019103932>

Kind regards,

Martina Rembold, PhD

Senior Editor

EMBO reports

Referee #1:

The manuscript is sound and the authors have clearly addressed my concerns as well concerns raised by the other reviewers. I find their experiments clear and appropriate for the comments they make. It is good that they have indicated the limitations in their conclusions and reworded them. The conclusions are in line with the experimental results and I recommend publication of this article. I do not find any issue with the language.

Referee #3:

The authors have sufficiently addressed my concerns. A considerable amount of new data has been added, which provides additional support for the conclusions of the authors. Publication is recommended.

All editorial and formatting issues were resolved by the authors.

Feng Bi
Sichuan University
37# Wai Nan Guoxue Road, Chengdu, Sichuan Province, 610041, P. R. China
China

Dear Dr. Bi,

I am very pleased to accept your manuscript for publication in the next available issue of EMBO reports. Thank you for your contribution to our journal.

Yours sincerely,
